

# Spectral Induced Polarization imaging to monitor seasonal and annual dynamics of frozen ground at a mountain permafrost site in the Italian Alps

Theresa Maierhofer[1,2], Adrian Flores Orozco[1], Nathalie Roser[1], Jonas K. Limbrock[4], Christin Hilbich[2],
Clemens Moser[1], Andreas Kemna[4], Elisabetta Drigo[5], Umberto Morra di Cella[3], Christian Hauck[2]

[1]Department of Geodesy and Geoinformation, TU-Wien, Vienna, 1040, Austria
[2]Department of Geosciences, University of Fribourg, Fribourg, 1700, Switzerland
[3]Environmental Protection Agency of Aosta Valley (ARPA), Saint-Christophe, 11020, Italy
[4]Institute of Geosciences, Geophysics Section, University of Bonn, Bonn, 53115, Germany
[5]Geologist freelance, Aosta valley, Saint-Pierre, 11010, Italy

*Correspondence to*: Theresa Maierhofer (theresa.maierhofer@tuwien.ac.at)

**Abstract**. We investigate the application of spectral induced polarization (SIP) monitoring to understand seasonal and annual variations of freezing and thawing processes in permafrost, in particular with regard to the frequency-dependence of the subsurface electrical properties. We installed a permanent SIP monitoring profile at a high mountain permafrost site in the
Italian Alps in 2019 and collected SIP data in the frequency range between 0.1-75 Hz over 3 years. Complementary seismic data were acquired, which, together with borehole data, were used to aid interpretation of the SIP imaging results. In particular, we investigated the phase frequency effect ($\phi FE$), i.e., the change of resistivity phase with frequency. We observe that this parameter ($\phi FE$) is strongly sensitive to temperature changes and might be used as a proxy to delineate spatial and temporal changes of ice content in the subsurface, providing information not accessible through electrical resistivity tomography (ERT)
or single-frequency IP measurements. Temporal changes in $\phi FE$ are validated through laboratory SIP measurements on samples from the site in controlled freeze-thaw experiments. We demonstrate that SIP is capable of resolving temporal changes in the thermal state and the ice/water ratio associated with seasonal freeze-thaw processes. We investigate the consistency between the $\phi FE$ observed in field data and ground water and ice content estimates derived from petrophysical modelling of ERT and seismic data.

## 1 Introduction

High-mountain environments are facing a rapid increase in air temperatures due to the amplification of the warming rate with elevation (Pepin et al., 2015) accelerating the rate of change of mountain permafrost systems and leading to increased permafrost temperatures, active layer thickening and decreasing ice contents (e.g., Biskaborn et al., 2019; Etzelmüller et al.
2020; Smith et al., 2022; Wu and Zhang, 2010). The change in the physical properties of the subsurface impacts slope stability



(e.g., Ravanel et al., 2017) and water management of water storage capacities and future water supplies (e.g. Harrington et al., 2018; Hilbich et al., 2022; Mathys et al., 2022; Rangecroft et al., 2016) making continuous temperature monitoring of the permafrost evolution in mountainous regions more and more important, i.e. as part of global (Global Terrestrial Network of Permafrost, GTNP), continental (Permafrost and Climate in Europe, PACE project) and regional (such as the Swiss Permafrost

Monitoring Network) monitoring programmes (e.g., Biskaborn et al., 2015; Harris et al., 2001; PERMOS, 2021).

Electrical resistivity tomography (ERT) is known to provide high-resolution spatial estimates about subsurface electrical properties sensitive to variations in temperature and water content, complementing spatially sparse borehole monitoring observations (e.g., Hauck, 2002; Hauck et al., 2013; Krautblatter et al., 2010; Oldenborger and LeBlanc 2018; Parkhomenko, 1982). Hence, in the last decades electrical monitoring arrays were set up at several permafrost sites throughout different

European mountain ranges to assess the temporal evolution of subsurface properties and processes relevant for permafrost investigations (e.g., Etzelmüller et al. 2020; Hauck, 2002; Hilbich et al, 2008; Isaksen et al., 2011; Keuschnig et al. 2017; Krautblatter & Hauck, 2007; Mollaret et al., 2019; Pogliotti et al., 2014; Supper et al., 2014). In unfrozen conditions, comparatively low resistivity values are observed due to electrolytic conduction taking place within the interconnected pore-space and along the electrical double layer (EDL) formed at the interface between water and mineral surfaces (e.g., Revil and

Glover, 1998; Ward, 1990; Waxman and Smits, 1968). Below the freezing point, the mobility of the ions is reduced, and parts of the liquid pore water is transformed into ice leading to an exponential increase of the electrical resistivity with decreasing temperatures (e.g., Hauck, 2002; Oldenborger and LeBlanc, 2018, Oldenborger, 2021).

Nonetheless, the interpretation of electrical signatures of ice, air and rock are still open to discussion and uncertainties remain (Hauck and Kneisel, 2008) that are often reduced by combining ERT investigations with complementary geophysical methods

such as Refraction Seismic Tomography (RST) due to its sensitivity to changes in mechanical properties between unfrozen and frozen materials (e.g., Hausmann et al., 2007; Hilbich, 2010; Mollaret et al., 2020; Steiner et al., 2021). Seismic and electrical data sets were combined in various studies to estimate volumetric ice, water and air contents using the so-called 4-phase model (4PM) approach (by Hauck et al., 2011). The 4PM has been implemented in a petrophysical joint inversion (PJI) framework for a simultaneous inversion of geophysical datasets (Wagner et al., 2019). However, the presence of ice and the

electrical double layer (EDL) formed at the water-ice interface requires the consideration of surface conduction (e.g., Bullemer and Riehl 1966; Carantl and Illlngworth 1982).

To account for surface conduction, the induced polarization (IP), also known as complex resistivity method has been tested for permafrost applications (e.g., Doetsch et al., 2015; Bazin et al., 2019; Duvillard et al., 2018, 2021). Expressed in terms of the complex resistivity, the real component relates to the resistivity measured through ERT, while the imaginary component

relates to surface conductivity arising from the accumulation and polarization of charges at the EDL formed in the ice-water or rock-water interface as well as protonic defects in ice surface. Moreover, the IP measurements collected at different frequencies (commonly between 0.1 to 1000 Hz in the so-called spectral IP, SIP) have revealed an improved delineation of the ground ice content as shown in Mudler et al. (2022). The enhanced polarization response at high frequencies is due to the motion of charge defects in the hydrogen-bonded ice lattice dominant at frequencies between 4 and 11 kHz for pure ice (Auty





and Cole, 1952). For lower frequencies, Kemna et al. (2014a) and Revil et al. (2019) attributed the SIP freezing/thawing behaviour in saturated rocks to both residual water films (bound water) at the ice and mineral surfaces, as well as the presence of water in smaller pores with a sufficiently reduced melting point as governed by the Gibbs-Thomson effect.

Duvillard et al. (2021) applied a petrophysical model calibrated through laboratory data to quantify the temperature distribution of a permafrost-affected rock ridge from time-domain IP data. Mudler et al. (2022) estimated the ice content obtained from

broad-band SIP inversion results conducted at a permafrost site in Siberia. Grimm and Stillman (2015) found temperature-dependent relationships between the ice volume fraction and the resistivity frequency effect (RFE) (see e.g. Vinegar and Waxman, 1984) in the laboratory and retrieved ice volumes at a frozen silt permafrost site. Maierhofer et al. (2022) showed that SIP data reduce the ambiguity in the interpretation of electrical resistivity data in a talus slope. They show that ice-rich blocky material, unfrozen coarse blocky environments and bedrock (unfrozen and frozen) can result in similarly high resistivity

values but different IP responses. Moreover, they demonstrate that ice-rich areas are associated with a significant increase in the IP response for data collected above 10 Hz, as also reported by Grimm and Stillman (2015).

Laboratory studies have investigated SIP responses in a frequency range between 0.01 Hz and 40 kHz during freezing/thawing periods observed on various types of porous media (e.g., sandstone, granite, soil and sands) within a temperature range varying between -15 °C and +20 °C (e.g., Coperey et al., 2019; Kemna et al., 2014a; Limbrock and Kemna, 2022; Olhoeft, 1977; Revil

et al., 2019; Stillman et al., 2010; Wu et al., 2017). However, SIP signals of natural permafrost soils and rocks during freeze-thaw cycles have rarely been explored. Doetsch et al. (2015) presented IP monitoring data acquired in the Arctic for a period of 4 months during freezing of the ground and hypothesize that changes in Cole-Cole parameters obtained from time-domain IP (TDIP) time-lapse inversions can reliably image freezing patterns. However, in their study, they did not measure the actual frequency-dependence of the IP response, but rather assumed a predefined relaxation model to explain the decay curve in TDIP

data. To our knowledge, there have been no studies investigating the changes in the frequency-dependence of the IP signatures (i.e., SIP signatures) due to subsurface seasonal freezing and thawing processes as well as processes on longer time-scales. Such investigations are important to extend petrophysical models such as the 4PM and PJI (1) to improve ice content estimations taking into account measured surface conductivity and (2) to account for spatially and temporally variable surface conductivity associated to varying ice-contents.

In this study, we apply the SIP imaging method in a high mountain permafrost terrain in the Italian Alps, where long-term borehole temperature and meteorological data are available for validation (Pogliotti et al., 2015). In particular, we investigate the frequency dependence of the signatures and imaging results covering a frequency range between 0.1 and 75 Hz for seasonal and annual variations along three years monitoring period. We hypothesize that SIP resolves temporal changes in the thermal state and the ice/water ratio of bedrock permafrost and the active layer (changes attributed to seasonal

freezing and thawing processes). We propose the use of the phase frequency effect (the difference in phase between high and low frequencies) as a proxy describing SIP responses without the necessity of fitting a relaxation model (e.g., the Cole-Cole). We demonstrate that such frequency-effect can be used to evaluate seasonal changes in measured and estimated volumetric



water content and unfrozen water content as well as PJI-derived ice content at one time instance. Laboratory results are used to sustain our field observations, in particular the phase frequency effect.

## 2 Material and Methods

### 2.1 Field site

The Cime Bianche monitoring site is located at the Cime Bianche Pass above Cervinia, Aosta Valley (Valtournenche Municipality), in the Italian Alps (Fig. 1) at an altitude of 3100 m a.s.l. (45° 55′09″N, 7° 41′34″E). The lithology of the Cime Bianche plateau is dominated by garnetiferous mica schists, calc-schists and amphibolites with a deeply weathered and fractured bedrock surface, thus, covering the bedrock with a layer of fine-grained to coarse blocky debris deposits with a thickness ranging from a few centimeters to a couple of meters (Pogliotti et al., 2015). Surface characteristics at the Cime Bianche site vary spatially, consist of soil with small parts covered with vegetation, silty parts, fine-grained calc-schists, small to coarse-grained amphibolite blocks and flat amphibolite bedrock outcrops and are thus typical of mountain permafrost environments. Gelifluction lobes and sorted polygons of fine material indicate the presence of permafrost, which was first recognized in 1990 (Guglielmin and Vannuzzo, 1995). Permafrost monitoring activities started in the late 1990s and the site was consecutively instrumented between 2004 and 2006 by the Environmental Protection Agency of Aosta Valley (ARPA VdA). Temperature measurements are conducted at different depths in a shallow borehole (SBH, reaching a depth of 7 m) and deep borehole (DBH, reaching a depth of 42 m) as well as a spatial grid of ground surface temperature loggers. An automatic weather station continuously monitors air temperature, soil moisture, net radiation, snow depth plus wind speed and direction (Pogliotti et al., 2015; Pellet et al., 2016).



**Figure 1: The Cervinia Cime Bianche study site located in the Italian Alps, with the SIP monitoring profile-, ERT/RST monitoring profile-, borehole- and meteorological sensor positions indicated. SIP data were collected along seven profiles: C1 denotes the long-term ERT/RST monitoring profile, SIPM the SIP monitoring profile and C1-C5 the SIP profiles collected for a spatial characterization of the site. © Google Maps**

120



In addition to the thermal and meteorological monitoring, annual ERT and RST (refraction seismic tomography) measurements are collected since 2013 to observe long-term spatio-temporal permafrost dynamics at the site (Pogliotti et al., 2015; Pellet et al., 2016; Mollaret et al., 2019). Mollaret et al. (2019) observed a significant warming in the boreholes and electrical resistivity decrease from August 2013 and 2017 along the entire ERT/RST monitoring profile with the strongest decrease in the area between the two boreholes. Ice contents along the ERT/RST monitoring profile (see Fig. 1) were estimated by Mollaret et al. (2020) using the PJI with maximum ice contents of around 25 %.

## 2.1 Ground temperatures and meteorological data

During the SIP monitoring period between September 2019 and September 2022, a mean annual air temperature (MAAT) of about -2.3 °C and mean annual precipitation of about 1200 mm yr-1 were measured at the site. Strong winds contribute to the spatially changing snow cover thickness at the site with less snow recorded at SBH compared to the DBH (Pogliotti et al., 2015), with a snow cover duration of ~260 days per year within the monitoring period. The borehole temperature data indicate that the permafrost is at least 42 m thick (max. depth of DBH, not shown in Fig. 2) with permafrost temperatures that vary between -0.5 and -1 °C (at 10 m depth, DBH) during the three years. Table 1 summarizes active layer thickness (ALT) evolution for the two boreholes with highest values measured in October 2020 and lowest values reported in October 2021. The observed spatial variability of the ALT between the two boreholes (i.e., ~2.5 m difference in ALT between SBH and DBH in 2021) can most likely be attributed to different ice/water contents due to different surface and subsurface conditions in terms of weathering and fracturing of the bedrock (Pogliotti et al., 2015).





**Figure 2: (a) Meteorological data (precipitation, snow depth, air and surface temperature), and ground volumetric water content measured at a depth of 20 cm and (b) borehole temperatures of the DBH and SBH for the SIP monitoring period between October 2019 and November 2022. (c) Temperature-depth plots for the respective time-lapse measurement dates for SBH and DBH.**



Seasonal variations in borehole temperature are consistent and correlate with air temperature, water content and snow height. From Fig. 2, it is seen that the surface was covered by snow for about 260 days a year and with larger average snow heights in winter 2019/20 (0.9 m) compared to 2020/21 (260 days, 0.5 m) and 2021/22 (190 days, 0.2 m). Due to the reduced insulating effect of relatively low snow height (e.g., Zhang, 2005), borehole temperatures reached lowest values in winter 2021/22. The highest air/surface temperatures were recorded in summer 2020, and volumetric water content (VWC) values observed during the summers 2020-22 were higher compared to summer 2019. Borehole temperatures were highest in summer 2022. During the monitoring period under investigation, the behavior of the VWC is characterized by a steady minimum throughout the winter (~9 %) followed by a strong increase at the beginning of the summer period (~ 20 %) due to snow melt (Fig 2 and Pellet et al., 2016). The highest variability of VWC can be observed during summer when precipitation sums are high.

**Table 1: Active layer thickness (ALT) evolution for the SIP monitoring period between October 2019 and August 2022**

| Year | ALT SBH (m) | ALT SBH (date) | ALT DBH (m) | ALT DBH (date) |
|------|-------------|----------------|-------------|----------------|
| 2019 | -5.9 | 2019-11-18 | -5.9 | 2019-11-13 |
| 2020 | -5.9 | 2020-10-20 | -6.7 | 2020-10-22 |
| 2021 | -3.5 | 2021-10-28 | -5.9 | 2021-10-29 |

## 2.3 Spectral induced polarization as monitoring method in frozen ground

### 2.3.1 The complex resistivity method

The induced polarization (IP) method, also referred to as complex resistivity (CR) method is a geophysical electrical technique which is based on measurements of the electrical impedance ($|Z|$) that can be performed in the time- or frequency domain. When conducted in the frequency domain (as in our study), two electrodes are used to inject a sinusoidal current into the ground and a second pair measures the resultant phase-shifted voltage. The impedance is then given as the amplitude ratio and phase-shift between the measured voltage and the injected current. When performed at different frequencies in the mHz-kHz range, commonly referred to as spectral IP (SIP), information on the frequency dependence of the electrical properties is gained. Multi-electrode measurements, in combination with inversion algorithms (e.g., Binley and Kemna, 2005), permit to resolve for variations of the complex electrical resistivity ($\rho^*$) (or its inverse, the complex electrical conductivity ($\sigma^*$)) in the subsurface (e.g., Kemna et al. 2012). The complex resistivity can be written as its magnitude ($|\rho(\omega)|$) ($\Omega$m) and phase shift ($\phi$) (rad) or expressed in terms of the real ($\rho'(\omega)$) ($\Omega$m) and imaginary ($\rho''(\omega)$) ($\Omega$m) components, such as (e.g., Binley and Slater, 2020; Wait, 1984):

$$\frac{1}{\sigma^*(\omega)} = \rho^*(\omega) = \rho'(\omega) - i\rho''(\omega) = |\rho^*(\omega)|e^{i\phi(\omega)}, \tag{1}$$

$$|\rho^*(\omega)| = \sqrt{\rho'(\omega)^2 + \rho''(\omega)^2} \text{ , and} \tag{2}$$



$$\phi(\omega) = \arctan(\frac{-\rho''(\omega)}{\rho'(\omega)}) , \tag{3}$$

where $\omega$ represents the angular frequency ($\omega = 2\pi f$, $f$ the excitation frequency, commonly between 0.1-1000 Hz) and $i$ is the imaginary unit ($i^2 = -1$). The real and imaginary components of the complex resistivity account for energy loss associated with conduction and energy storage associated with polarization (capacitive property). In natural media, with a negligible amount of electronic conductors, conduction takes place in the pore fluid through the migration of ions (electrolytic conduction, $\sigma_{el}$) and within the electrical double layer (EDL) present at the interface between grains and electrolyte (surface conduction, $\sigma'_s$) (e.g. Schwarz, 1962; Leroy et al., 2008). The EDL results from the attraction of counterions by charged mineral surfaces, depending on surface charge and surface area. In the presence of an external electric field, charges in the EDL polarize resulting in the polarization effect (measured by the imaginary component of complex conductivity, $\sigma''_s$). Accordingly, for frequencies < 1 kHz, the complex conductivity $\sigma^*$ (or $\rho^*$)) can be written as (e.g. Binley and Slater, 2020)

$$\sigma^*(\omega) = \left(\sigma_{el} + \sigma'_s(\omega)\right) + i\sigma''_s(\omega). \tag{4}$$

In the measured frequency band its frequency dependence can often be described by the Cole-Cole model (Cole and Cole, 1941).

Considering that field-SIP devices are often limited in the frequency range and might lack the possibility to recover the frequency maximum required to fit a dispersion model such as the Cole-Cole, we propose here to quantify the frequency-dependence in the SIP signatures using the phase frequency-effect ($\phi FE$). Such parameter represents the difference between the low frequency and high frequency $\phi$ values over a spread of frequencies ($A$) according to

$$\phi FE = \frac{log_{10}(-\phi(A\omega_0)) - log_{10}(-\phi(\omega_0))}{log_{10}(A\omega_0) - log_{10}(\omega_0)}, \tag{5}$$

where $\omega_0$ is the lowest frequency (in Hz) and the product $A\omega$ represents the maximum frequency investigated (in Hz). Grimm and Stillman (2015) proposed a similar analysis but they only took into account the change in electrical resistivity over a spread of frequencies and ignored the polarization parameters ($\phi$ and $\sigma''$).

### 2.3.2 Temperature dependence of electrical properties

The dependence of rock electrical properties on temperature differs for temperatures above and below the freezing point. Above the freezing temperature, in the presence of liquid pore water in a metal-free porous rock, the bulk electrical conductivity is controlled by the electrolytic conductivity of the liquid pore water and the surface conductivity, while the polarization effect for frequencies below 1 kHz are mainly related to electrochemical (ionic) polarization in the EDL at the pore water - solid matrix interface (Kemna et al., 2012). The temperature dependence of the electrical conductivity is controlled by the temperature dependence of the ionic mobility which is inversely related to the viscosity of the fluid and can be approximated by a linear relationship (Revil and Glover, 1998, 2012; Zisser et al., 2010):

$$\sigma(T) = \sigma(T_0)[1 + \alpha_T(T - T_0)], \tag{6}$$



where $\alpha_T$ is the temperature compensation factor for a fluid that depends on the choice of the reference temperature $T_0$ and $T$ is the temperature at which $\sigma$ is measured. Experimental $\alpha_T$ values lie typically around $0.02\,°C^{-1}$ at 25 °C for most electrolytes, while recent studies (Oldenborger, 2021) derived different $\alpha_T$ values of Eq. 7 (i.e., $\alpha_T = 0.02\,°C^{-1}$, $T_0 = 20\,°C$; $\alpha_T = 0.0254\,°C^{-1}$, $T_0 = 10\,°C$; $\alpha_T = 0.0321\,°C^{-1}$, $T_0 = 0\,°C$). Binley et al. (2010) measured the change in real and
imaginary conductivity with positive temperature and reported an approximate increase of $\sigma'$ and $\sigma''$ (at 1.5 Hz) with temperature of 2 % per °C using Eq. (7). Other empirical formulations modelled the temperature dependence of electrolytic conduction by an Arrhenius-fit (e.g. Oldenborger, 2021).

When water changes its phase from liquid to solid, the salinity of the liquid pore water phase increases (e.g., Hobbs, 1974; McKenzie et al., 2007). Above the eutectic temperature (i.e., the temperature where water starts to crystallize) and below the
freezing temperature, ionic exclusion leads to remaining salt within the pore water (e.g., Coperey et al., 2019; Oldenborger and LeBlanc, 2018; Revil et al., 2019) and an increase in fluid conductivity. During ice formation, the decrease in water content with temperature is gradual and follows a freezing curve (or thawing curve), where no super-cooling is present (e.g., McKenzie et al., 2007). The saline unfrozen water is composed of residual water films (i.e., bound water) at the ice and mineral surfaces and water in smaller pores with a reduced melting point related to capillary and adsorptive forces according to the Gibbs-
Thomson effect (see, e.g., Dash et al., 2006; Kemna et al., 2014a; Mohammed et al., 2018). Duvillard et al. (2018) and Coperey et al. (2019) extended the Stern-layer model of Revil (2012, 2013b, 2013a) to freezing conditions to describe the dependence of the electrical conductivity and the normalized chargeability on temperature below 0 °C including parameters as the residual water content, porosity, freezing and characteristic temperature. Similar resistivity-temperature models were used to account for the temperature dependence of the electrical resistivity upon freezing for geophysical investigations in permafrost terrain
(Oldenborger and LeBlanc 2018; Oldenborger, 2021).

### 2.3.3 Temperature correction and prediction of unfrozen water content and volumetric water content from petrophysical relationships

The distribution of subsurface electrical resistivity depends on soil properties such as saturation ($S_w$), fluid resistivity ($\rho_w$) or conductivity ($\sigma_w$), lithologic properties and surface conduction ($\sigma_s'$) (e.g., Lesmes and Frye, 2001), which can be written as:
$$\sigma_b = S_w^n \phi^m \sigma_w + \sigma_s' \tag{7}$$

where $m$ is defined as the cementation exponent and $n$ is the saturation exponent. While the saturation exponent varies in the range $n = 2 \pm \frac{1}{2}$, the cementation exponent can range from 1 (for rocks with a low porosity but well-developed fracture network) to around 5 (for rocks with a low connectedness of the pore and fracture network) (Glover, 2009).

Neglecting surface conduction, Oldenborger (2021) computes the volumetric water content as a function of porosity and
saturation ($S_w$), such as:
$$VWC = (S_w \cdot \phi) \cdot 100, \tag{8}$$

with saturation being resolved from ERT data using the relationship, which can be written as:



$$S_w = e^{\frac{ln\left(\frac{\rho_{corr}}{\rho_w \cdot \phi^{-m}}\right)}{n}}.\qquad(9)$$

Within our study, we used the same value (i.e., $100\ \Omega m$) for the pore water resistivity as previous studies at the site (Mollaret

et al., 2020) and estimated the cementation exponent for a given measurement of VWC, with a constant factor $a = 1$ and

assuming $n = 2$. In Eq. 9, the $\rho_{corr}$ represents a temperature correction applied to the bulk electrical resistivity data at 0.5 Hz.

Such correction is based on the model by Oldenborger (2021), which can be written as:

$$\rho_{corr} = \rho[1 + \alpha_T(T - T_0) + \beta_T(T - T_0)^2],\qquad(10)$$

where $\beta_T$ is an additional temperature compensation coefficient.

Upon freezing, $S_w$ decreases and $\rho_w$ increases due to ionic exclusion and estimates of the unfrozen water content $S_w^F$, which is

the fraction of water remaining unfrozen even at relatively low temperatures, can be derived by (Daniels et al., 1976)

$$S_w^{F(1-n)} = \frac{\rho_F}{\rho_0}\qquad,\qquad(11)$$

with $\rho_F$ being the the resistivity for frozen and $\rho_0$ for unfrozen material for which it is assumed that the pore space was

completely saturated prior to freezing ($S_w = 1$). Using an exponential relationship for subfreezing temperatures,

$$\rho = \rho_0 e^{b(T_F - T)},\qquad(12)$$

Hauck (2002) and Oldenborger and LeBlanc (2018) derived estimates of the unfrozen water content as follows

$$S_w^F = e^{\frac{b(T_F - T)}{1-n}}\qquad,\qquad(13)$$

with the constant $b$ being a function of depth.

### 2.3.4 Monitoring set-up and acquisition of SIP imaging data

An array of 64 stainless steel electrodes with a separation of 3 m between electrodes connected with coaxial (shielded) cables,

resulting in a profile length of 189 m was permanently installed in October 2019 at the Cime Bianche plateau. The position of

the SIP monitoring (SIPM) profile was chosen away from boreholes (c.f. Figure 1) and other metallic structures to avoid

interferences due to electronic conduction. The position of the profile aimed at covering different surface characteristics,

namely fine-grained, coarse-blocky areas and parts dominated by bedrock outcrops found along the study area towards the

understanding of spatio-temporal SIP imaging results associated to different substrates during the monitoring period.

Measurements were conducted with an eight-channel data acquisition system (DAS-1 manufactured by Multi-Phase

Technologies). The configuration protocols consisted of dipole-dipole (DD) and multiple gradient (MG) arrays and deployed

coaxial cables to reduce cross-talking (e.g., Flores Orozco et al., 2013). During the monitoring period, between October 2019

and August 2022, SIP data were collected with 12 different frequencies in the range between 0.1-225 Hz using a DD

configuration with dipole lengths of 12 m (skip-3) as normal and reciprocal (N&R) pairs for the quantification of data error

(e.g., LaBrecque et al., 1996; Flores Orozco et al., 2012; Slater et al., 2000) yielding a total number of 1474 quadrupoles (Table

A, Appendix). The MG configuration was deployed with a total number of 744 quadrupoles using 4 potential dipoles (with

lengths of 3 m, 6 m, 9 m and 12 m) within the current dipole in order to collect data sets with a higher signal-to-noise ratio



(e.g., Dahlin & Zhou 2006) compared to the DD configuration. These MG surveys were especially important during the winter
period (between November and June), related to poor galvanic contact between electrodes and the ground (also called contact
resistance $R_s$), thus deteriorating the quality of the DD measurements.

$R_s$ were recorded before each data acquisition with observed mean values between 20-160 kΩ and mean current injections ($I$)
varying between 0.2-15 mA (Table A, Appendix). The highest $R_s$ values were measured in the coarse-blocky part with
observed values of 30-90 kΩ in the summer period and 125-190 kΩ in the winter months. Lowest values were acquired for
electrodes located in the fine-grained part of the profile $R_s$ lying between 5-20 kΩ in summer and between 40-90 kΩ in winter.
During the SIP monitoring period, we avoided artificially wetting the electrodes for a reduction of $R_s$ (e.g., Mollaret et al.,
2019; Maierhofer et al., 2022), to ensure comparable conditions throughout the year (as proposed by e.g. Hilbich et al., 2011).
We aimed at collecting an entire SIP data set once a month during a period of 2 years (between October 2019 and October
2021), to capture seasonal and annual changes in the polarization response. However, inaccessibility to the study area during
the Corona pandemic and technical problems lead to gaps in the SIP monitoring data in December 2019 and December 2020,
as well as to the period between March and June 2020. In addition to the monthly measurements, we collected another end-of-
summer dataset in August 2022.

### 2.3.5 Data filtering, data error analysis and inversion procedure of SIP monitoring data

We processed DD and MG data for every frequency and monitoring date independently, following the automated processing
procedure presented in Fig. 3. For all DD data sets, we removed as erroneous readings those related to $R_s > 200$ kΩ (are
considered as open circuit) and/or negative impedance magnitude ($Z$) values, as well as positive impedance phase values ($\phi$).
In case of DD measurements, outliers were identified (and removed) as those quadrupoles for which the misfit between normal
and reciprocal impedance magnitude ($\Delta Z$) and phase ($\Delta \phi$) readings was larger than 50% of the average value ($Z_{NR}, \phi_{NR}$) and
larger than twice the standard deviation of the N&R misfit of the entire data set, similar to Maierhofer et al. (2022). For MG
surveys, where no reciprocals were measured, we developed an analysis scheme for data quality that considers the relationship
between phase and voltage readings to assess signal strength and a moving average filter to remove spatial outliers and
erroneous measurements (c.f., Fig 3). In other words, we removed isolated readings based on plots from phase readings plotted
against their associated voltages. The resulting numbers of data after filtering are summarised in Table A (Appendix) for each
survey.

After the analysis of the independent data sets, we analysed the temporal evolution of the number of quadrupoles for each
frequency. This is needed to warranty that imaging results obtained from different times are related to similar resolution. Due
to the high number of outliers detected for winter measurements (up to 80% for 75 Hz), we did not systematically remove
unrepeated quadrupoles for the entire monitoring period. We performed analysis for detailed processes: (1) the thawing of the
active layer during the summer period (June to October 2020), and (2) the inter-annual changes based on summer
measurements (August 2020, 2021 and 2022). For those two specific analyses, only repeated quadrupoles were used for
inversion.



Analyses of the misfit between N&R readings were used for the quantification of data error (corresponding to small fluctuations between N&R) in the inversion, as described in previous studies (e.g., LaBrecque et al., 1996; Slater et al., 2000). Similar to the study of Maierhofer et al. (2022), we used the same error parameters for the inversion of different frequencies to fit the spectral data to the same error level. The analyses of the N&R misfit revealed distinct variations in the error parameters for data sets collected in summer and winter. Hence, we performed the inversion of each SIP dataset with mean error values for each time instance with the same error parameters used for DD and MG data, since error models allow the use of the same error parameters for configurations where no reciprocals are available.

For the inversion of the SIP data, we used the finite-element smoothness-constraint inversion code CRTomo (Kemna, 2000), which computes the distribution of the complex resistivity in the subsurface from a given data set of impedance magnitude and phase values at a distinct frequency. Data-error estimates are defined in the inversion and permit to fit the observed data within its noise level, which is defined by the error model (Kemna, 2000). The iteration is stopped if the RMS difference between model response and data reaches a value below its target value of 1, after which smoothing is again increased until the RMS value is equal to 1 (within a tolerance interval). Thus, the inversions converge to the target value fitting the observed data within their estimated noise level for the smoothest possible model. Inversion results presented here were computed independently for every data set collected at different frequencies and times.

To assess the reliability of the monitoring inversion results, especially regarding the resolved electrical properties at depth, we make use of the depth of investigation (DOI) index, introduced by Oldenburg and Li (1999). In our analysis, we calculated the DOI index for resistivity and phase images for all frequencies as

$$R(x,z) = \frac{m_1(x,z) - m_2(x,z)}{m_{1r} - m_{2r}} \tag{14}$$

with constant reference models of $m_1 = 1000\ \Omega\mathrm{m}, -5\ \mathrm{mrad}$ and $m_2 = 10\,000\ \Omega\mathrm{m}, -75\ \mathrm{mrad}$. We chose the values of the reference models as mean minimum and maximum resistivity and phase values observed in all SIP images. We blank model regions with large DOI values (i.e., $> 0.2$ as proposed by Oldenburg and Li, 1999) in all imaging results presented in this study.



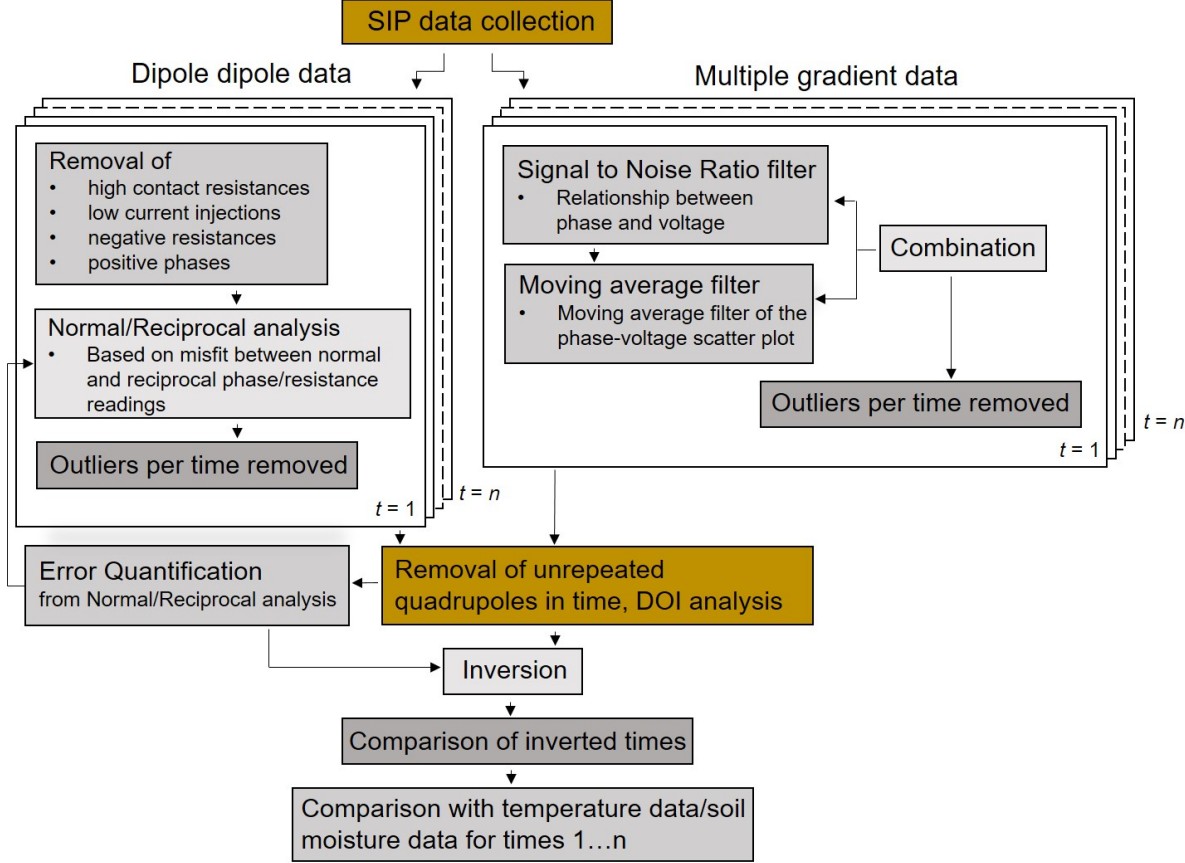

**Figure 3: Work flow of SIP data processing for the monitoring data set at Cime Bianche, Cervinia.**

## 2.4 Complementary data: 3D electrical survey

In September 2019, we collected ERT data along five profiles C1, C2, C3, C4 and C5 (Fig. 1) to investigate the spatial heterogeneities of the active layer and subsurface characteristics at the monitoring site. The data were measured in 12 frequencies between 0.1 and 225 Hz using the DAS-1 device with a DD configuration with dipole lengths of 12 m (skip-3). For profiles C2, C3 and C5, 32 electrodes and a separation of 4 m was chosen, whereas profiles C1 (prolongation of the ERT/RST monitoring profile) and C4 consist of 64 electrodes and 2 m separation. Data quality was evaluated by means of N&R analysis and the inversion of the resistivity data at 0.5 Hz was performed in 3D using the modules of the open-source library pyGIMLi (Rücker et al., 2017). We used mean error parameters ($a$=0.1 $\Omega$, $b$=5 %) from N&R analysis for the inversion of the ERT data sets to fit all data to the same error level enabling a better comparison of the multi-dimensional (different profiles and frequencies) inversion results similar to Lesparre et al. (2017).



## 2.5 Complementary data: Refraction seismic tomography and petrophysical joint inversion

Refraction seismic tomography (RST) measurements were conducted along the SIP monitoring profile in August 2020 with a 24-channel Geode instrument (manufactured by Geometrics) and 24 geophones (corner frequency of 30 Hz) with a roll-along of 12-geophones overlap. In total, 36 receiver and 14 shot locations were used with ~15 individual shots recorded between electrode 12 and 48 (between profile meter 33 to 141) keeping the same geophone spacing as for electrodes. We used a sledgehammer as source of the seismic waves with hit points between every second geophone along the seismic refraction line. The individual shots were stacked ~10 times to improve the signal-to-noise ratio of the seismic data. First-break travel time picking was manually done using the open-source library formikoj (Steiner and Flores Orozco, 2023) which provides modelling, reading and basic processing of seismic waveform data. A 120 Hz low-pass filter was applied to the seismic traces to mitigate the influence of high-frequency noise. For the independent inversion of the picked P-wave travel times we used the modules of the open-source library pyGIMLi with an estimated RST data error of 2.2 ms from comparing a subset of travel times that was picked twice.

We used the PJI framework developed by Wagner et al. (2019) to quantify the four volumetric fractions of ice ($f_i$), water ($f_w$), air ($f_a$) and rock matrix ($f_r$), based on the filtered apparent resistivity values and picked P-wave travel times. We used the same values as previous studies at the site (Mollaret et al., 2020). Namely, the initial porosity was set to 40 %, constituent velocities ($v_p$) as 1500 m/s for pore water, 3750 m/s for ice, 4000 m/s for the rock matrix and 330 m/s for air. We accounted for surface conductivity in addition to electrolytic conduction by introducing a constant surface conductivity term in Archie's law. We used a surface conductivity range between 0.001 and 0.01 S/m (based on laboratory studies on pure ice (Bullemer and Riehl 1966; Carantl and Illlngworth 1982) and rock samples under freezing conditions (Coperey et al., 2019)) and chose a final constant value across the imaging plane of 0.01 S/m as it resulted in the lowest chi-square. The final PJI results are presented in Appendix B. A relative error of 5 % was used for the ERT data following the N&R analysis and an absolute error of 2.2 ms for the picked travel times in the inversion, which resulted in an error-weighted chi-square fit of 1.0.

## 2.6 Complementary data: SIP laboratory data

Laboratory measurements were performed on a cylindrical solid rock sample of 10 cm length and 3 cm diameter of amphibolite (with a porosity of 3.5 %) and a loose sediment surface sample, both collected close to the SIP monitoring profile. The electrical impedance was measured in a frequency range of 10 mHz to 45 kHz during controlled freeze-thaw cycles (+20°C to -40°C to +20°C), using a 4-channel SIP-04 impedance spectrometer (Zimmermann et al., 2008). The solid rock sample was fully saturated with water of similar fluid conductivity as in field conditions (0.01 S/m), sealed in a shrinking tube, and placed in a climate chamber where it was cooled down from +20°C to -40°C by successively changing the temperature in steps of 0.2°C - 4°C with a duration of 120 to 180 minutes for each temperature step. Limbrock and Kemna (2022) found such settings to be sufficient for the given size and thermal properties of the sample to reach thermal equilibrium. Afterwards, the samples were heated up again following the same temperature procedure. For potential measurements, we used ring electrodes made out of



tinned copper with a distance of 3 cm. The loose sediment sample was saturated up to a volumetric water content of about
365 20%, with the same type of water and placed in a cylindrical container with a total length of 18 cm, a diameter of 4 cm, and a
potential electrode separation of 6 cm. Due to the increased sample size, the duration of each temperature step was increased
by 60 minutes.

## 3 Results

### 3.1 Site characterization

We present in Fig. 4 the 3D electrical resistivity results as slices extracted at different depths parallel to the surface obtained
from the 3D inversion of the five ERT profiles (C1-C5) collected in September 2019. Fig. 4 shows lower electrical resistivity
values in the shallow subsurface (between 0 and 4 m depth) than for deeper layers (> 5 m depth), with slightly lower values
observed in the western compared to the eastern part of the study area. The thickness of the low-resistive surface layer
(inflection point in electrical resistivity) is slightly overestimated (~4 m depth at SBH and ~5 m depth at DBH) compared to
375 the thaw depth measured in the boreholes at the date of the geophysical measurements (3 m in the SBH and 4 m in the DBH)
most likely due to the smoothing in the inversion. Low electrical resistivity values (< 8000 Ωm) are observed in the northern
and the central-western part of the study area for all depths; whereas the highest electrical resistivity values are observed at >
5 m depth in the central, central-eastern and central-southern parts of the study area. Between 6 and 20 m depth we observe no
strong resistivity changes in depth associated to the permafrost body.







**Figure 4: Spatial characterization of the Cime Bianche monitoring site for the 3D electrical resistivity inversion results from five profiles collected in September 2020 (dotted black lines). Panels a)-f) represent as slices parallel to the surface in six different depths. The borehole location of the SBH and the DBH are marked. The coordinates of the SIP monitoring profile are shown in addition (dotted grey line), but these data were not used for inversion due to a differing measurement date (October 2020).**

The baseline SIP imaging results of October 2019 are presented in Fig. 5 expressed in terms of the phase ($\phi$), the real ($\rho'$) and imaginary ($\rho''$) components of the CR, obtained for data collected along the SIP monitoring profile for low (0.5 Hz) and high (75 Hz) frequencies. While the frequency dependence is pronounced in the polarization parameters (for $\rho''$ and $\phi$), we observe no variation in $\rho'$ with increasing frequency for the whole imaging plane. Similar to the results shown in Maierhofer et al. (2022), absolute $\rho''$ and $\phi$ values are higher for 75 Hz (hereafter $\rho_{75}''$ and $\phi_{75}$) than for 0.5 Hz ($\rho_{0.5}''$ and $\phi_{0.5}$). In general, comparatively low $\rho'$, $\rho''$ and $\phi$ values ($\rho' \sim 3000 - 10\,000\ \Omega m$, $\rho_{0.5}'' \sim 10 - 100\ \Omega m$, $\phi_{0.5} \sim 0 - 10$ mrad, $\rho_{75}'' \sim 100 - 1000\ \Omega m$, $\phi_{75} \sim 10 - 100$ mrad) are resolved at depths < 4 m for $\rho$, $\rho_{75}''$, and $\phi_{75}$ and < 5 m for the polarization at 0.5 Hz, which can be related to the thaw layer (with a thaw depth of 4 m for both boreholes at the measurement date) composed of



fine- to coarse-grained material (see the surface characteristics shown in Fig. 5a). It is underlain by a high-resistivity layer with high absolute polarization values for 0.5 and 75 Hz ($\rho' \sim 10\,000 - 30000\ \Omega\mathrm{m}$, $\phi_{0.5} \sim 10 - 30$ mrad, $\phi_{75} \sim 100 - 300$ mrad), which is representative for the permafrost body. The contrast between the two layers is highest for $\rho'$ and $\phi_{75}$.

Concerning lateral changes in the SIP images, the highest electrical resistivity and absolute phase values are observed for all depths between profile meter 138 and 162 corresponding to the coarsest amphibolite blocks of 1-2 m size found at the surface

of the SIP monitoring profile (see Fig. 5a). The uppermost 1-3 m of profile meter 93 to 117 reveal the lowest resistivity values and absolute phase values at 75 Hz ($\rho' \sim 3000\ \Omega\mathrm{m}$, $\phi_{75} \sim 100$ mrad) along the profile, which can be related to a thin layer of silty soil with some vegetation and fine-grained calc-schists observed at the surface (see Fig. 5a). The contrast to the frozen bedrock is clearly visible for this part of the profile for all components of the complex resistivity except for $\rho_{75}''$ and $\phi_{75}$, where we resolve a low-polarizable anomaly also at depth ($\rho_{75}'' < 1000\ \Omega\mathrm{m}$, $\phi_{75} < 60$ mrad). In the northern part of the

monitoring profile with intermediate grain size (between 0 and 51 m), we observe low real resistivities and absolute phase values at the surface ($\rho' \sim 4000 - 6500\ \Omega\mathrm{m}$, $\rho_{0.5}'' \sim 40 - 70\ \Omega\mathrm{m}$, $\phi_{0.5} \sim 5$ mrad, $\rho_{75}'' \sim 200 - 500\ \Omega\mathrm{m}$, $\phi_{75} \sim 40 - 70$ mrad) and at depth ($\rho' \sim 6500 - 15\,000\ \Omega\mathrm{m}$, $\rho_{0.5}'' \sim 70 - 140\ \Omega\mathrm{m}$, $\phi_{0.5} \sim 5 - 15$ mrad, $\rho_{75}'' \sim 500 - 1600\ \Omega\mathrm{m}$) for 0.5 Hz and intermediate absolute phase values for 75 Hz ($\phi_{75} \sim 15 - 200$ mrad).




(a)

big, flat bedrock outcrops
(amphibolite), soil and small
blocks of amphibolite

small blocks of amphibolite,
soil, gravel, debris cover
thickness (> 2m)

fine-grained
calc schist, soil
and vegetation
(layer thickness
of ~50 cm)

small
blocks of
amphibolite
and silty
parts

coarse blocks of
amphibolite
blocksize (1-2m)

small blocks of
amphibolite
(metabasite),
vegetation

(b)

$\rho'$ at 0.5 Hz

$\rho''$ at 0.5 Hz

$\phi$ at 0.5 Hz

$\rho'$ at 75 Hz

$\rho''$ at 75 Hz

$\phi$ at 75 Hz




**Figure 5: SIP baseline imaging results, corresponding to data collected in October 2019. Panel a) represents different substrate classes, which are discussed in more detail in the Discussion Section. (b) Baseline complex resistivity model (October 2019) at two frequencies (0.5 Hz and 75 Hz) with the two boreholes located at the center of the profile (the SBH 20 m away, the DBH 50 m away), and the black layer representing the thaw depth at the measurement date.**


### 3.2 The temperature dependence of the SIP method

Fig. 5 reveals clear variations in the polarization for different depths (i.e., temperatures) and frequencies. To better understand the temperature dependence of the polarization response, we show in Fig. 6a, the measured impedance field data (apparent resistivity magnitude ($\rho_{app}$) and phase ($\phi_{app}$)) in a frequency range of 1-75 Hz between October 2019 and November 2021

including warm (i.e., summer months, from July to October) and cold (i.e., winter months, from November to May) periods. For comparison, Fig. 6b shows the laboratory measurements in the same frequency range of 1-75 Hz, corresponding to a thawing cycle, with varying temperatures between -10 °C and +10 °C.

In Fig. 6a, the $\rho_{app}$ values differ by ~14 $k\Omega$m between summer and winter months with no frequency-dependence observed. Similarly, the apparent phase shift clearly shows higher absolute values in winter ($\phi_{75} \approx 300$ mrad, $\phi_1 \approx 10$ mrad) compared

to summer months ($\phi_{75} \approx 50$ mrad, $\phi_1 \approx 15$ mrad). Additionally, Fig. 6a shows a general increase in the absolute $\phi_{app}$ values with increasing acquisition frequency above 7.5 Hz (inflection point between low and high response), with a more pronounced increase in winter (ca. 290 mrad) than in summer (ca. 35 mrad). These variations in $\rho_{app}$ and $\phi_{app}$ between summer and winter are linked to seasonal temperature variations between +10 and -8 °C measured in the borehole close to the subsurface.

Similar to the field data, we observe also for the SIP laboratory measurements a clear distinction in the polarization response

between unfrozen and frozen states. Absolute phase values are low for positive temperatures and show no significant frequency-dependence; whereas the absolute $\phi_{app}$ values increase both with frequency and with decreasing temperature. This increase in absolute phase during freezing starts at a frequency of ~10 Hz for the loose sediment sample and at ~50 Hz for the solid rock sample. Thus, the inflection point seems to change depending on the texture/ice content of the samples analyzed in the laboratory. Such effect can be related to the well-known temperature-dependent relaxation behaviour of ice (as shown for

pure ice by Auty and Cole, 1952). Additionally, at temperatures < 0 °C, we observe a higher increase in absolute $\phi_{app}$ values between 1 Hz and 70 Hz for the loose sediment sample (45 mrad at -10 °C) compared to the solid rock sample (15 mrad at -10 °C) suggesting an increase in absolute $\phi_{app}$ values with increasing ice content. Accordingly, Fig. 6 suggests that the frequency-dependence in the phase measurements might be used as proxy to sense the response of ice, i.e., to discriminate between regions with no ice and regions richer in ice, as also observed in Maierhofer et al. (2022). Clearly, the frequency range

in the SIP data needs to extend above this sensitive frequency, which corresponds to an inflection point between the low response (likely due to the geology) and the high values associated to ice.



When comparing the SIP monitoring and laboratory data, we find a similar behaviour in the polarization during freezing and thawing cycles, with an up to eight times higher increase in absolute $\phi_{app}$ between 75 and 1 Hz for frozen compared to unfrozen state. However, the increase in absolute phase is smaller for laboratory data (2-15/5-45 mrad for the rock/sediment sample) than for field data (35-290 mrad) in the temperature range covered by both data sets (-8 to +10°C observed within the active layer at our field site).



**Figure 6: (a) Temporal changes of SIP measurements between October 2019 and November 2021 (frequency range: 0.5-75 Hz). Impedance magnitude expressed in terms of its phase and magnitude (converted to apparent resistivity) values and shown for level**



**1 and 2 (i.e. 3 and 6 m electrode separation between current and potential dipoles, which corresponds to the active layer) at a horizontal distance of 80 m. The SIP data are presented for all DD monitoring data sets with different symbols and colors representing different years and months, respectively. (b) SIP laboratory measurements on a solid rock sample and a loose sediment**
**sample collected at the site close to the monitoring profile. Apparent resistivity and phase values are shown in a frequency range between 1 Hz and 75 Hz for temperatures between -10 °C and +10 °C.**

### 3.3 The phase frequency effect and its link to temperature, water content and ice content estimates

Figure 6a shows that our SIP measurements at the field scale do not solve for the peak in the frequency-dependence but extend
above the inflection point associated to a sensitive frequency, where phase values increase with increasing frequency and ice content. Hence, we propose to analyze the phase frequency effect ($\phi FE$) presented in Equation 5. To evaluate the applicability of such parameter, we present in Fig. 7 the lab measurements collected during (7a) freezing and (7b) thawing cycles in terms of apparent resistivity ($\rho_{app}$ at 0.5 Hz) and for the $\phi FE$ (calculated between 70 and 0.5 Hz). After saturating and cooling both samples from +10 °C to ~0 °C, both $\rho_{app}$ and $\phi FE$ follow a linear trend, with values increasing with decreasing temperature.
During initial freezing, supercooled conditions with constant $\rho_{app}$ and $\phi FE$ values occur between 0 °C and -2.1 °C for the loose sediment sample, and between 0 °C and -2.8 °C for the solid rock sample. A sudden increase in $\rho_{app}$ (average rate of change of 3 $k\Omega m/°C$ for the sediment sample, 1.9 $k\Omega m/°C$ for the rock sample) and $\phi FE$ (average rate of change of 0.044 per °C for the sediment sample, 0.031 per °C for the rock sample) is observed below this temperature as well as an exponential increase in the electrical parameters ($\phi FE$ and $\rho_{app}$) with decreasing temperature. Such effect is indicative of abrupt ice
formation in the pores of the samples. During thawing from -10°C to -0.1 °C (for the loose sediment sample) respectively -0.7 °C (for the solid rock sample), $\rho_{app}$ and $\phi FE$ gradually decrease with a smaller average rate of change for the rock sample (1.4 $k\Omega m/°C$, 0.021 $per °C$) than for the sediment sample (3 $k\Omega m/°C$, 0.03 $per °C$), suggesting continuous melting of ice in the pore space. The $\phi FE$ slightly increases and decreases between -0.1 °C and 0.5 °C for the loose sediment sample and between -0.7 °C and -0.2 °C for the solid rock sample before following a linear trend as during freezing. During thawing, $\rho_{app}$
and $\phi FE$ are generally higher than during freezing ($\Delta\phi FE = 0.05$). Similar to earlier studies (Kemna et al., 2014a; Krautblatter et al., 2010), a lowering of the freezing point of water in smaller pores of the solid rock sample is observed due to capillary and adsorptive forces according to the Gibbs-Thomson effect. At subfreezing temperatures, we observe overall higher $\rho_{app}$ and $\phi FE$ values for the loose sediment sample compared to the solid rock sample, which can be explained by a higher ice content of the loose sediment sample compared to the solid rock sample.





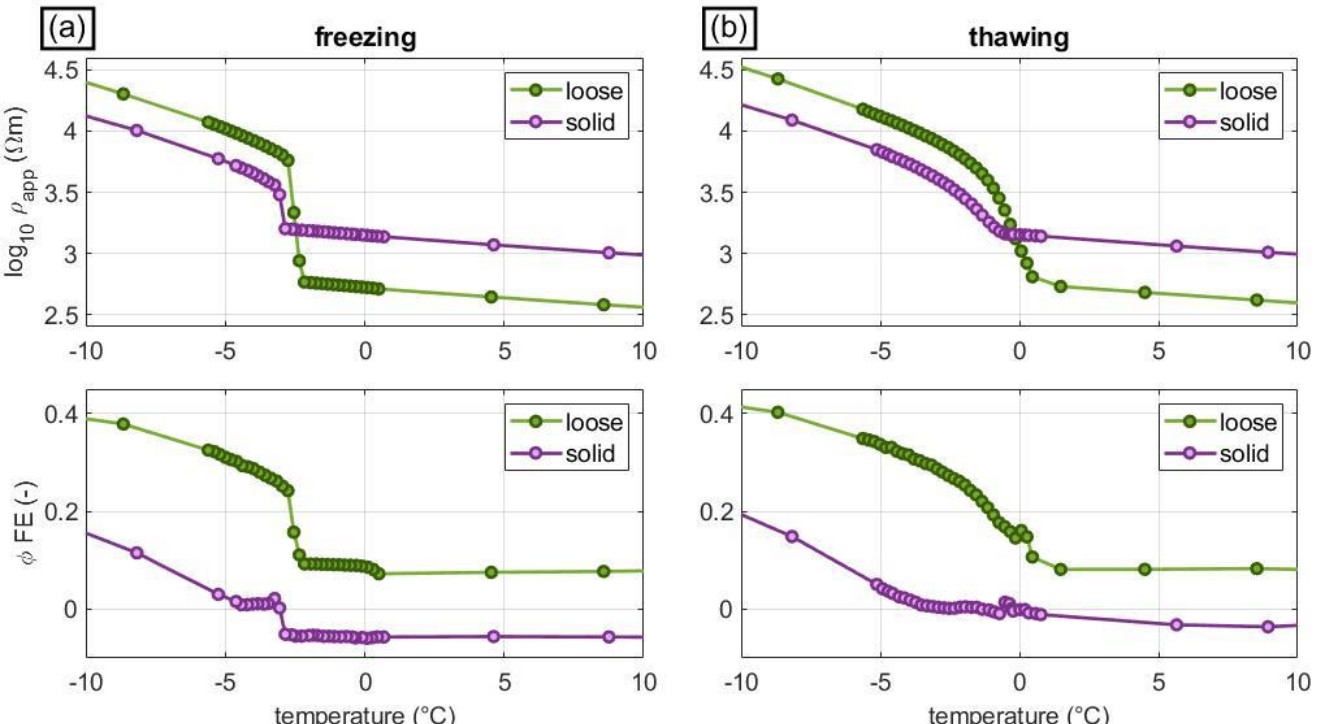

**Figure 7: Temperature dependence of the impedance magnitude and the phase frequency effect measured on solid rock (solid) and loose sediment (loose) samples during (a) freezing and (b) thawing in the laboratory (same laboratory experiments as shown in Fig. 6).**


Fig. 8 shows the spatial variation in the $\phi FE$ computed from the inversion of field data collected in August 2020 along the SIP monitoring profile. To investigate a possible correlation with ice and water content, we compare this image with PJI-estimated ice and water contents ($f_i$, $f_w$) resolved from ERT and RST data collected on the same day in the same profile. The complete inversion results of the PJI parameters (electrical resistivity, seismic velocities as well as rock and air contents) are presented

in Appendix B.

The $\phi FE$ in Fig. 8c ranges from 0.2 to 0.4, similar to the values observed in the laboratory, and is highest at 5-10 m depth (~ 0.35 at the borehole location) and lowest in the thaw layer (< 0.3 at the borehole location). Fig. 8a-b reveal that PJI-estimated $f_i$ values are minimal (but differ by ~7-10 % from zero) and $f_w$ reaches its maximum (10-33 %) in the thaw layer. Highest $f_i$ (~ 22 % at the borehole location) and lowest $f_w$ (~ 4 % at the borehole location) are encountered between 5 to 10 m depth. The

comparison of vertical 1D logs of $\phi FE$, $f_i$ (Fig. 8d) and $f_w$ (Fig. 8e) at the borehole location suggests a high and positive linear correlation between $\phi FE$ and $f_i$ ($r^2 = 0.9$, Fig. 8f), and a lower and negative correlation between $\phi FE$ and $f_w$ ($r^2 = 0.5$, Fig. 8g). In general, there is a good correlation between low $\phi FE$ and low $f_i$, and high $f_w$; while high $\phi FE$ correspond also with high $f_i$, and low $f_w$, supporting our hypothesis that the phase frequency effect can be used as a proxy for the ice content – and indirectly temperature and water content. The main inconsistency between the three parameters can be observed between 90



and 115 m, where a low-$\phi FE$ anomaly coincides with high $f_i$. Within this part of the profile, we observe at the surface the material with the finest texture (i.e., silty soil, box 2 in subpanel (5a)) with calc schists visible at the surface. Hence, the polarization parameter might be able to resolve a decrease in ice content that the PJI is not able to resolve due to the lack of sensitivity of electrical resistivity and seismic velocities to textural properties.

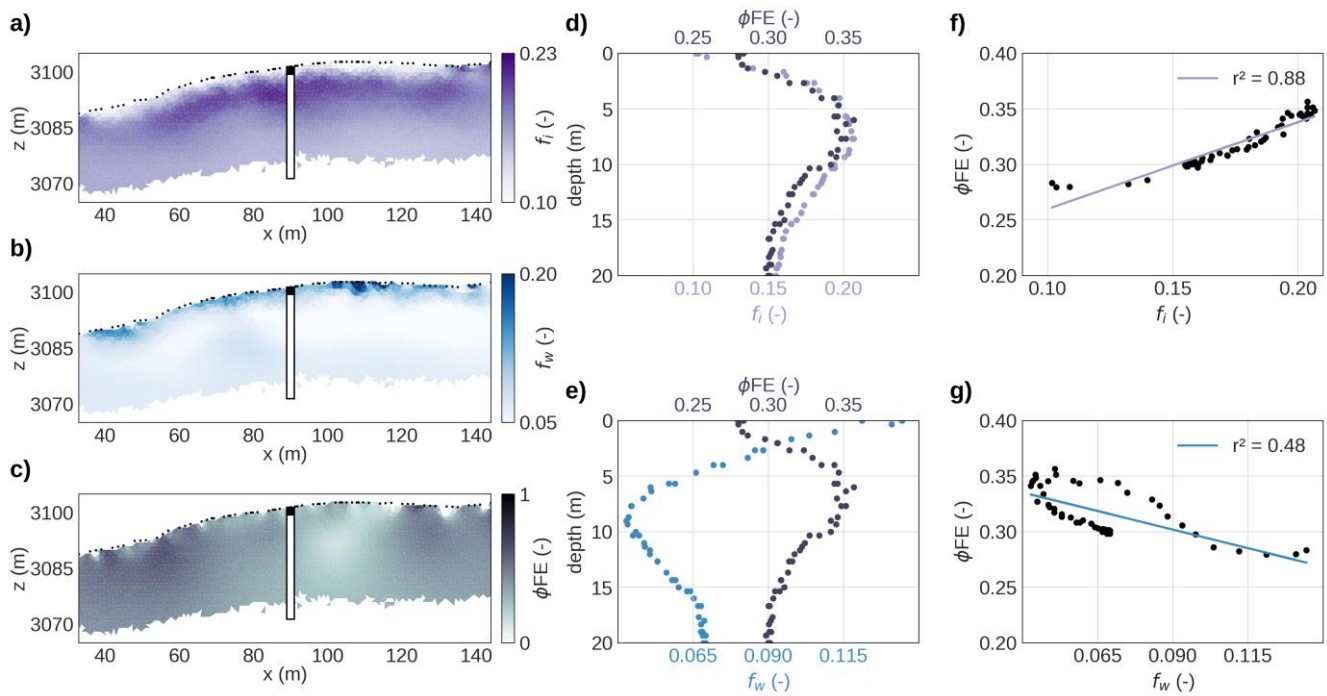

**Figure 8: (a-c) Calculated $\phi FE$ for the phase of the complex resistivity imaging results between 75 Hz and 0.5 Hz and visualized for the whole tomogram for the SIP monitoring profile (between profile meter 33 and 141) and comparison with water and ice content estimates from PJI results. (d) Vertical 1D logs of the $\phi FE$ and the ice content and (e) water content at the borehole location. The correlations between the two (volumetric) phase fractions and the $\phi FE$ are shown in (f) and (g).**

### 3.4 Temporal changes in the SIP results

In Figure 9, we show the temporal evolution in borehole temperatures and the complex resistivity at different frequencies (0.5, 7.5, 25 and 75 Hz) extracted for the borehole location at different depths. During the 2-years-monitoring period, we observe that complex resistivity results mainly follow the thermal evolution, but with distinct differences between the two years. Freezing started in both years (2019 and 2020) in November, with the advance of the freezing front reflected in a decrease in temperature and an accelerated increase of the electrical parameters. The winter 2020/21 was much colder than 2019/20, which is represented by ~ two times higher $\rho'$, similar $\phi_{0.5}$ and more than three times higher $\Delta\phi_{75}$ values in January 2021 compared to January 2020. Borehole temperatures and $\rho'$ reached their minimum and maximum values, respectively, in March 2021



$(T = -5\,°C, \rho' = 6.8\,k\Omega m)$, while absolute $\phi$ values were highest already in January 2021 ($\phi_{0.5} = 60$ mrad, $\phi_{7.5} = 180$
mrad, $\phi_{25} = 450$ mrad, $\phi_{75} = 2000$ mrad). During spring 2021 (April-June), a steady increase in shallow borehole
temperatures lead to seasonal ice melt in the uppermost layer and a small decrease in shallow electrical parameters. An abrupt
increase in subsurface temperatures at the end of June 2021 is reflected in a decrease of 30 % in $\rho'$, a decrease of 85 % in $\phi_{0.5}$
and a decrease of 90 % in $\phi_{75}$. Between March and June 2020, no SIP measurements were conducted. During summer 2021,
we observe smaller shallow absolute $\phi$ values compared to the summer period in 2020, which can be explained by a higher
VWC observed in summer 2021 than in summer 2020 (i.e., Fig. 2).



**Figure 9: Comparison of borehole temperatures with complex resistivity results at different frequencies (0.5, 7.5, 25, 75 Hz) extracted at borehole position in the borehole temperature sensor depths.**



## 3.5 Complex resistivity – temperature relationship

Fig. 10 presents the complex resistivity ($\rho^*$) data extracted from inversion results along the SIPM profile at the borehole location for 4 frequencies and a comparison to the borehole temperatures ($T$). The $T$-$\rho^*$ relationship is shown for two material compositions: debris (Fig. 10a) and bedrock (Fig. 10b) with the transition between debris and bedrock at the borehole location estimated at a depth of 1.2 m (for further details see Appendix B). The $T$-$\rho^*$ was fitted for each frequency using a linear model for positive temperatures and an exponential model for subzero temperatures with the coefficients of determination obtained for the two regions and different frequencies summarized in Table C1 of Appendix C.

For unfrozen conditions, we observe that $\rho'$ and absolute $\phi$ values are similar for debris and bedrock and show a linear increase in both parameters with decreasing but still positive temperature. The increase is higher for $\rho'$ compared to $\phi$. Additionally, $\rho'$ and $\phi$ values are higher at 75 Hz than at 0.5 Hz (with a difference between them of 700 Ωm and 50 mrad).

Below the freezing point and for both material compositions, $\rho'$ and absolute $\phi$ increase exponentially with decreasing temperatures, with the highest absolute $\phi$ and lower $\rho'$ values observed at 75 Hz compared to 0.5 Hz. The frequency effect in $\phi$ at -1 °C is different for debris and bedrock with a pronounced increase in $\phi$ for the debris (310 mrad) resolved for data at 0.5 and 75 Hz. Such high $\phi FE$ reflects the expected high ice contents in winter periods in the small to coarse blocks found at the surface. Accordingly, we observe a modest change in the phase values (70 mrad) for the bedrock, where we have lower porosity and, thus, also lower ice contents. These results coincide with the observations made in the laboratory (Fig. 7), where the $\phi FE$ is higher at subfreezing temperatures for the loose sediment sample compared to the solid rock sample. Additionally, we observe a lower freezing/melting point for the bedrock compared to the debris layer, which is related to the pore size according to the Gibbs-Thomson effect (e.g., Dash et al., 2006; Kemna et al., 2014a) and was also observed in the laboratory for the solid rock and sediment sample.

In the freeze-thaw transition processes, hysteresis effects are often observed also at the field-scale as shown by Mollaret et al. (2019) and Wu et al. (2013). In Fig. 10c we present the seasonal cycle at shallow depths (within the debris layer), where freezing and thawing processes can be distinguished in $\rho'$ and $\phi$ at 7.5 Hz (which represents the sensitive frequency) and for the $\phi FE$ calculated between 0.5 and 75 Hz. Similar to the laboratory results, we also observe at the field-scale lower $\rho'$, $\phi$ and $\phi FE$ during freezing (from October to January) than during thawing (from May to July) which is related to the difference in liquid water for the two processes. Upon freezing, supercooling leads to a remaining unfrozen pore fluid below the freezing point until ice starts to form, thus, resulting in a larger amount of unfrozen water compared to thawing processes at the same temperature (Wu et al., 2017). Additionally, the freezing point is lowered due to ions being excluded from ice formation and accumulating in the liquid phase (Bittelli et al., 2004), similar to what was observed in the laboratory. The supercooling effect is also reflected in Fig. 10c with lower $\rho'$ and $\phi$ values observed at 7.5 Hz in October/November, where subsurface temperatures decreased down to -2.2 °C. Only in January-February, an increase in $\rho'$, $\phi$ and $\phi FE$ occurred with peak values detected between February and May. During snow melt in May-June, the $\rho'$, $\phi$ and $\phi FE$ values drop abruptly with the absolute phase at 7.5 Hz decreasing from 100 to 10 mrad and the $\phi FE$ falling below 0.25 as soon as water content increases. The zero-





curtain effect could not be observed during the phase change as the temporal resolution of the monitoring was too low during
this period.



**Figure 10: Temperature dependence of the complex resistivity at different frequencies (0.5. 7.5, 25 and 75 Hz) obtained from 2-years SIP monitoring data extracted at borehole location for all temperature sensor depths down to 20 m. The temperature dependence**
**of the real component and the phase of the complex resistivity are shown for the (a) debris (first 1.2 meters of the subsurface) and (b) bedrock. The complex resistivity data were fitted for each frequency using a linear model for positive temperatures and an exponential model for subzero temperatures according to equations C1-C4 of Appendix C. In (c) we show the temperature-resistivity relation, temperature-phase relation at 7.5 Hz and the temperature-phase frequency effect relation for all available dates.**

## 4 Discussion

### 4.1 Temporal variability of the phase frequency effect and unfrozen water content

We investigated the subsurface polarization properties for both seasonal and inter-annual changes along a transect located in

a mountain permafrost environment in the Italian Alps. Our monitoring results reveal an overall increase in resistivity and



phase of the complex resistivity during the winter months. However, such an increase is also observed at larger depths (i.e.,
below 10 m, within the permafrost layer) where expected changes in temperature, water content and ice content are small.
High phase readings in winter are on the one hand due to low current densities injected in frozen terrain covered by snow,
resulting in low signal-to-noise-ratios (S/N). On the other hand, further distortions in the IP data are due to parasitic
electromagnetic fields (i.e., capacitive coupling) in the measurements above 1 Hz arising from large variations in electrode
contact resistance, especially for large dipole lengths (Binley et al. 2005; Flores Orozco et al., 2018; Ingeman-Nielsen, 2006;
Kemna et al. 2012; Zimmerman et al., 2008). The N&R analysis revealed a significant increase in the error parameters for the
winter measurements of 30-50 % demonstrating lower S/N in winter and higher uncertainties in the imaging results obtained
for measurements between January and May.

Different SIP monitoring studies (e.g., Lesparre et al. 2017, Flores Orozco et al., 2019) use the same error parameters for the
inversion of the entire monitoring period assuming that this permits the comparison of imaging results obtained with the same
level of contrasts, i.e., sensitivity. However, such an approach assumes fairly consistent S/N across the monitoring data set.
Accordingly, within our analysis, assuming the same error parameters for the entire monitoring period lead to either under-
fitting the readings in summer (assuming the error level from winter measurements), or over-fitting the winter data when
inverted to the error parameters observed in summer measurements. Hence, we define error parameters for each data set
collected at different times and frequencies, which also considers the frequency-dependence observed in our data (similar to
Flores Orozco et al., 2011).

Our results suggest a correlation between ice content and the increase in the frequency-dependence of the polarization, in
particular the resistivity phase values. Hence, we propose the use of the so-called phase frequency-effect ($\phi FE$) as a proxy for
the interpretation of SIP imaging results in terms of ground ice content. Frolov (1973) showed that the electrical properties of
the frozen soil are determined by the specific surface area of the soil, the ice and unfrozen water content. Figure 11a illustrates
how the $\phi FE$ changes as a function of time and depth at the borehole location with a comparison to (11b) volumetric water
content (VWC) and (11c) unfrozen water content (UWC) variations during freeze-thaw processes.

As discussed above, the $\phi FE$ results in winter show up to 3 times higher values than those for the summer period due to the
high data uncertainty in winter measurements. Nonetheless, freeze-thaw transitions are well-resolved and reveal consistent
seasonal trends for the parameters $\phi FE$, VWC and UWC as observed in Fig 11. During the freezing period (from October to
January) we observe lower $\phi FE$ than during thawing (from May to July), which is related to the difference in water content
for the two processes. As discussed by Bittelli et al. (2004) and Wu et al. (2017), upon freezing, supercooling leads to a
remaining unfrozen pore fluid below 0 °C and a decreased freezing point due to ionic exclusion and accumulation within the
liquid phase. This results in a higher unfrozen/volumetric water content and lower $\phi FE$ compared with thawing processes at
the same temperature ($\Delta VWC \approx 6\ \%, \Delta\phi FE \approx 0.15$). During the freezing period in 2019, measured VWC is high until January
2020 (measured VWC=18%), while in the subsequent years water contents drop to their yearly minimum (VWC=8%) already
in November. The $\phi FE$ follows the same trend. The drop in water content due to ice formation is always accompanied by an
increase in the $\phi FE$. Similar to our study, Wu et al. (2017) observed an increase in phase at 1 Hz between -2 and -4 °C in



saline permafrost, which they relate to supercooling effects and the initiation of ice nucleation. In spring (beginning of July 2020/ end of June 2021), the shallow $\phi FE$ values decrease abruptly ($\Delta\phi FE \sim 0.17$) as soon as VWC increases to its maximum (17-30%). During the summer period we observe a steady decrease in water content and increase in phase frequency effect, with increasing air and subsurface temperatures.

Although measured VWC and those derived from $\rho'$ show similar trends and lie in the same range, we observe wetter conditions at the beginning of snow melt which only affect the measured VWC. This can be attributed to the different volumes of investigation from electrical measurements and the in-situ sensor, as well as possible inaccuracies in the calibration of the petrophysical model. Additionally, water content estimates can be improved by taking into account surface conductivity from SIP studies. Eq. 8 used to calculate VWC neglects surface conduction; however, this cannot be neglected in the presence of ice within the subsurface, as we observe an increase in polarization ($\rho''$ and $\phi$) in our laboratory and field data. We note here that the imaginary conductivity is considered a direct measure of surface conductivity (e.g., Binley and Slater, 2020); thus, our parameters $\rho''$ and $\phi$ can also be used as approximations to understand the spatio-temporal variations in surface conductivity. Duvillard et al. (2018) performed a 3D-survey at a rock glacier in France and analyzed the slope of the trend in normalized chargeability as a function of conductivity data, with the authors suggesting that the ratio between $R = 8 \cdot 10^{-2}$ and $R = 12 \cdot 10^{-2}$ indicates that surface conduction dominates over other mechanisms. However, such relationship focuses only on the low-frequency response of the signatures and does not take into account the surface conductivity of ice. Our results reveal a significant frequency-dependence of the polarization, especially between the low (< 7.5 Hz) and high frequencies, which we quantify through the $\phi FE$. Accordingly, we cannot apply the equation proposed for the temperature dependence in freezing conditions by Duvillard et al. (2021) and assume that electrolytic conduction dominates in our data, especially for measurements above 7.5 Hz. In case of time-domain measurements, such as those conducted by Duvillard et al. (2018), the parameter retrieved is the normalized chargeability, which is an average value that accounts for the polarization over the frequency-bandwidth recovered through the sampling of the decay-curve. Accordingly, samplings in the time-domain as collected in the studies of Duvillard et al. (2018) and Duvillard et al. (2021) with long pulses (1 s and 1.5 s) will relate rather to processes taking place at solid-fluid EDL (as observed in our case <7.5 Hz) and not to the polarization of ice observed in our lab and field measurements at higher frequencies. Hence, SIP data and the use of the $\phi FE$ could therefore help to assess changes in the surface conductivity accompanying temperature changes and to delineate spatial changes in water and ice content in monitoring and mapping applications as demonstrated above and in the study by Maierhofer et al. (2022).





**Figure 11: Temporal changes in $\phi FE$, volumetric water content and unfrozen water content at 99 m horizontal distance (borehole location) along the monitoring profile and at various depths (0.6 - 10 m). The cementation exponent was estimated for a given measurement of VWC (baseline date in October 2019) within the debris layer ($\phi = 0.6$). Porosity, fluid resistivity, cementation exponent and saturation exponent were kept constant over time.**

## 4.2 Spatio-temporal variability of the phase frequency effect

Pogliotti et al. (2015) found a pronounced spatial variability in the active layer thickness at the two boreholes in Cervinia Cime Bianche and argue that the difference can be related to contrasting water contents and varying surface and subsurface conditions in terms of weathering and fracturing of the bedrock (see table 1 for ALT during the monitoring period 2019-2022). They suggest a higher ice content in the SBH compared to the DBH, which was further supported by PJI results of Mollaret et



al. (2020) with different ice contents observed at both borehole locations (up to 10% at the DBH, 17% at the SBH). Additionally, our results show (c.f., Fig. 4) an overall shallower uppermost layer in the northeast of the study site (1-2 m difference to southwest) with lower $\rho'$ at depth close to the DBH compared to the SBH.

We have observed a relationship between the $\phi FE$ and the ice-content estimated by PJI (Fig. 7), as well as the consistent temporal changes in $\phi FE$ (Fig. 11) at different periods during the monitoring, but this analysis has only been conducted at the borehole position, where additional ground temperature data are available for validation. To evaluate the applicability of the proposed parameter for spatio-temporal changes across the entire imaging plane, Fig. 12a shows the resolved $\phi FE$ along the SIPM profile and the computed annual changes in $\phi FE$ between August 2020 and August 2022 (Fig. 12b), as well as the

comparison of $\phi FE$ values extracted at the borehole position (Fig. 12d) with borehole temperature readings of the SBH and the DBH (Fig. 12c). Both, the distribution with depth as well as the interannual differences show a consistent pattern between $\phi FE$ and borehole temperatures. At the borehole location, we observe for all three dates lower $\phi FE$ ($< 0.35$) in the unfrozen active layer compared to zones at $> 5$ m depth related to the permafrost body with $\phi FE$ values ranging from 0.35 to 0.45. From numerical modelling (not shown) we found that an electrode separation of 3 m with dipole lengths of 3 and 12 m were

small enough to delineate the transition between the thawed active layer and the permafrost using the $\phi FE$. Previous geophysical investigations along the ERT/RST monitoring profile in August 2017 report low seismic P-wave velocities of $< 1000\ ms^{-1}$ and low electrical resistivities of $< 10\ k\Omega m$ within the uppermost 4-5 m thick layer and intermediate electrical resistivities ($10 - 50\ k\Omega m$) and high seismic velocities ($2000 - 5000\ ms^{-1}$) for the permafrost zone below (Mollaret et al., 2020). PJI results of the seismic and electrical data yielded ice contents of ~15 % within the permafrost body at the SBH,

where also the $\phi FE$ shows high values. Figure 12a also reveals clearly lower $\phi FE$ in the permafrost at depths between 8 and 15 m in 2020 ($\phi FE \approx 0.36$) compared to the colder year 2022 with $\phi FE$ values in the range of 0.45 (see yellow box in Fig. 12a). This annual increase in $\phi FE$ is consistent with borehole temperature data of the DBH, which show a cooling at 8 m depth between 2020 and 2022. Hence, our results suggest that a yearly temperature change of $\approx$0.2 °C at depth can be resolved by the change in the $\phi FE$ of 0.05, suggesting a high sensitivity of SIPM to small changes in temperature that might not be

resolved through ERT monitoring alone. Below 20 m, we assign low credibility to the $\phi FE$ values due to large DOI values (i.e.,$> 0.2$) and low sensitivity in the computed images.

The general tendency of $\phi FE$ for the entire imaging plane of the SIPM profile is similar to that of the borehole location and thus, consistent with temperature changes, with positive $\phi FE$ differences in the active layer and negative differences at depth (Fig. 12b and 12c). However, discrepancies can be observed for some parts of the profile. In the northern part of the SIPM

profile (i.e., between 0-30 m), with lowest $\rho'$ values and small $\phi FE$ for all dates, also $\phi FE$ changes are the smallest during the monitoring period, with a slight decrease in $\phi FE$ ($\phi FE$ difference $\approx$ -0.1) between August 2020 and August 2021 and an increase between 2021 and 2022 ($\phi FE$ difference ~ 0.1). Due to the low $\phi FE$ and $\rho'$ values at all depths in the north of the study site (see, e.g. Fig. 4), we interpret this part of the monitoring profile as an area of low ice contents. The highest $\rho'$ and $\phi FE$ values are observed in the south between 140- and 189- meters profile length, where the surface is dominated by the



coarsest blocks within the study site. Here, we find a consistent increase in $\phi FE$ over the 2 years with positive $\phi FE$ differences of ≈0.3 between 2020 and 2022 indicating a high ice content within this part of the SIPM profile. Between profile meter 40 and 60 as well as between 90 and 115 m, two anomalies are resolved, characterized by not consistent changes in the $\phi FE$ during the monitoring period. The high $\phi FE$ anomaly between 40-60 meters corresponds to high ice contents and low porosity resolved from PJI (c.f. Fig. 7). At the position of the second anomaly, the surface cover is fine-grained and consists of silt, soil

and vegetation and revealed the highest water content, as well as small $\rho'$ and $\phi FE$ values. While the permafrost body is generally assumed to be completely frozen for consecutive years with only small ice content changes at depth, strong spatio-temporal dynamics in the $\phi FE$ at depth need to be considered with caution, as the data are inverted independently for measurements at different frequencies and time-lapses, and thus, may be subject to uncertainties associated to inversions with different regularization, sensitivity and contrasts. Such uncertainties could be reduced by the simultaneous inversion of SIP

data measured over a range of frequencies (e.g. Günther and Martin, 2016; Kemna et al., 2014b; Son et al., 2007) and the inclusion of a spatio-temporal constraint. Yet, the small changes in temperature (0.2 K) observed in the borehole, demonstrate that there is a change at depth, which seems to be captured with the $\phi FE$. Whether the magnitude of the $\phi FE$ is biased by the smoothness constraint inversion, is something that needs to be further explored in future studies.








**Figure 12: (a)** The phase frequency effect ( $\phi FE$ ) and **(b)** the difference in $\phi FE$ (i.e., $\phi FE\ difference = \phi FE\ Aug22 - \phi FE\ Aug20$) is visualized between August 2020 and August 2022 for the entire imaging plane. **(c)** Borehole temperature data of SBH and DBH and the **(d)** phase frequency effect extracted at the borehole position for the dates of the respective SIP measurements of the 3 years are compared. The position of the borehole within the SIP profile is marked with a vertical line and the respective thaw depth of the baseline measurement date is highlighted.




**4.3 Comparison of the phase frequency effect and PJI ice and water content estimations**

Our analysis based on field and laboratory data suggests that temperature, water and ice contents are dominant factors controlling the frequency dependence of the SIP data throughout the freeze-thaw transition and show a high potential for an improvement in the estimation of these volumetric fractions within the subsurface. The frequency-dependent electrical properties of ice have been studied in the laboratory for pure ice, as well as for frozen solid rock and sediment samples (e.g. Auty and Cole, 1952; Bittelli et al., 2004; Grimm et al., 2010; Limbrock and Kemna, 2022) and at the field-scale (e.g. Grimm and Stillman, 2015; Mudler et al., 2022) revealing the temperature-dependent relaxation behaviour of ice (Auty and Cole, 1552) at higher frequencies (with the ice relaxation peak frequency observed in the range between 1 and 45 kHz). By fitting a two-component Cole-Cole model to the SIP data, either for single laboratory measurements (Limbrock and Kemna, 2022) or within a 2-D inversion for field data, Mudler et al. (2022) separated the different polarization responses and estimated subsurface ice content using an existing two-component complex resistivity model of frozen soil considering the ice relaxation as the dominant process. However, such broadband spectral induced polarization measurement device (i.e., the Chameleon-2 used in the study of Mudler et al., 2022) is still not suited for monitoring and cannot be used to perform an extensive mapping as presented here. Considering that field-SIP devices are often limited in the frequency range and do not capture the ice relaxation peak, we cannot use petrophysical models linking relaxation parameters and ice content nor the fitting of Cole-Cole parameters. Instead, we propose the $\phi FE$, which has been revealed consistent to other data such as temperature and water content changes as discussed above. Hence, the approach presented here and the $\phi FE$ parameter suggested might be applicable to other monitoring studies and surveys conducted with electrical devices available at the market but limited in the frequency range (for example the LIPPMANN and the Geotom measurement devices, which can collect IP measurements between 0.2 and 30 Hz). Ice content has not been directly quantified at the study area, hindering a direct comparison between $\phi FE$ and this key parameter. The only information about the ice-content at the site has been presented by Mollaret et al. (2020) through PJI modelling of data collected in the ERT/RST profile. We have seen that at the borehole location (see Fig. 8), $\phi FE$ and PJI-estimated ice content are strongly correlated. In this regard, the $\phi FE$ could be an initial way to incorporate SIP information into the 4PM, and improve the estimation of ice-content in PJI.

Fig. 13 evaluates the correlation between the $\phi FE$ and the ice and water contents estimated from the PJI for selected zones of interest (ZOIs) in different parts of the tomogram and for depths representative for the active layer and the permafrost body. The results show that a specific relationship, such as observed at the borehole location, is not observed in all regions of the tomogram. High $\phi FE$ values in the permafrost body mostly coincide with high ice contents and low water contents and vice versa; however, it is not observed in all ZOIs, for instance within the part of fine-grained material at the surface corresponding to ZOI4. Additionally, $\phi FE$ fluctuations within the active layer are high and $\phi FE$ values are in the range of the permafrost body for some of these ZOIs in the active layer. Such variations in the active layer could be minimized in the future by regularization strategies as for example the minimal gradient support (Blacheck et al., 2008), which permits to solve for sharp



contrasts between interfaces, and still warranty smooth variations within the different zones. Also, further studies should investigate the regularizations across time and space dimensions to enhance the consistency between the resolved electrical models and improve the quantitative meaning of the $\phi FE$.

Differences observed in Fig. 13 between the $\phi FE$ and PJI-derived ice contents at depth might indicate that ice contents might be overestimated in some areas due to the anisotropic smoothness constraint inversion or the insensitivity of SRT and ERT to

possible fractures, so that lateral changes in ice content are not resolved. As mentioned above, the PJI is strongly sensitive to the porosity model defined as a start model for the inversion (see, e.g. Steiner et al., 2021 for a discussion) and the gradient model forces a lateral smoothness in the parameters, which might not be present at the site. Such lateral variations will be better solved in the SIP imaging and likely better retrieved in the frequency dependence. Additionally, Maierhofer et al. (2022) demonstrated that surface conduction changes spatially, with large polarization ($\sigma''$) associated to ice-rich areas (even at low

frequencies, due to EDL polarization at the ice-water interface) and also our results (see Fig. 5 and Fig. 9) reveal that surface conductivity changes not only spatially, but also temporally, with $\rho''$ and $\phi$ changing across different positions of the profile (geological changes) and different seasons. Hence, as ice content changes across the site and over time, spatially and temporally varying surface conduction needs to be considered additionally to electrolytic conduction within the PJI to obtain reliable petrophysical parameters.

Figure 13 also evidences an over-estimation of the ice-content in the active layer through the PJI. The occurrence of non-zero ice content estimates in regions where they are unlikely, such as in areas where temperatures are positive, have also been reported by Pellet et al. (2016) and Wagner et al. (2019), who proposed and implemented physical constraints on water content (based on soil moisture measurements) or temperature-dependent constraints on ice occurrence during inversion. However, such ice constraints are limited to sparsely distributed point locations and might bias the constraints used in the inversions, for

those areas without information. We can see in Fig. 13 that the minimal $f_i$ is about 10%, even for the ZOI retrieved from the AL. This problem can be resolved by fixing a spatial constraint with an ice-content of 0% in the active layer, yet such an inversion would require exact information about the geometry of the active layer. Hence, the ice-content resolved through the PJI cannot be used directly to evaluate the proposed $\phi FE$.

The quantitative interpretation of $\phi FE$ is likely still open to debate; however, its applicability is supported in our study through

the analysis of independent measurements in well-controlled experiments conducted in the laboratory (Fig. 6 and Fig. 7). These analyses reveal a clear distinction in the polarization response between unfrozen and frozen states and a significant frequency-dependence in the absolute $\phi$ values with decreasing temperature, which in turn is related to the well-known temperature-dependent relaxation behaviour of ice (Auty and Cole, 1952). Similar to our analysis, also Limbrock and Kemna (2022) report about the temperature-dependent relaxation behaviour of ice for the SIP response of all samples analyzed in the laboratory

with variations in shape and strength for different textures and mineralogy of the samples. This increase in absolute phase during freezing starts at a frequency of ~10 Hz for the loose sediment sample and at ~50 Hz for the solid rock sample. Laboratory and field results reveal that between 5 and 8 Hz an inflection point between the low polarization response at low frequencies (likely associated to the polarization of grain-water interfaces) and the increase in the response at high frequencies





exist due to the polarization of ice. Such an inflection point depends on the texture of the rock as well as the ice content (and

temperature) as evidenced in our laboratory measurements. Additionally, at temperatures $< 0$ °C, a higher increase in absolute

$\phi$ values for the loose sediment sample (in the laboratory) and the debris (at the field) compared to the solid rock sample

(laboratory) and the bedrock (field) suggests a correlation of $\phi$ with ice contents and the possibility to use the frequency-

dependence in the phase measurements as proxy to sense the response of ice, i.e., to discriminate between regions with no ice

and regions richer in ice, as also observed in Maierhofer et al. (2022) and Mudler et al. (2022). Clearly, the SIP measurements

need to symmetrically capture the response at frequencies above and below the inflection point where the response is

dominated by the high values due to the polarization of ice. We also note here that the application of the $\phi FE$ needs to be

conducted for measurements at other sites, corresponding to different lithology, porosity and ice content. Moreover, other

polarization processes occurring at higher frequencies, for example in the presence of high clay content, metallic minerals or

peat within the subsurface, would have also an influence on the $\phi FE$ and need to be addressed in future studies. In this regard,

the $\phi FE$ offers so far rather qualitative information about the geometry and temporal changes in areas rich in ice. Further

studies might attempt to include such information at least as a spatial-constraint within the PJI.



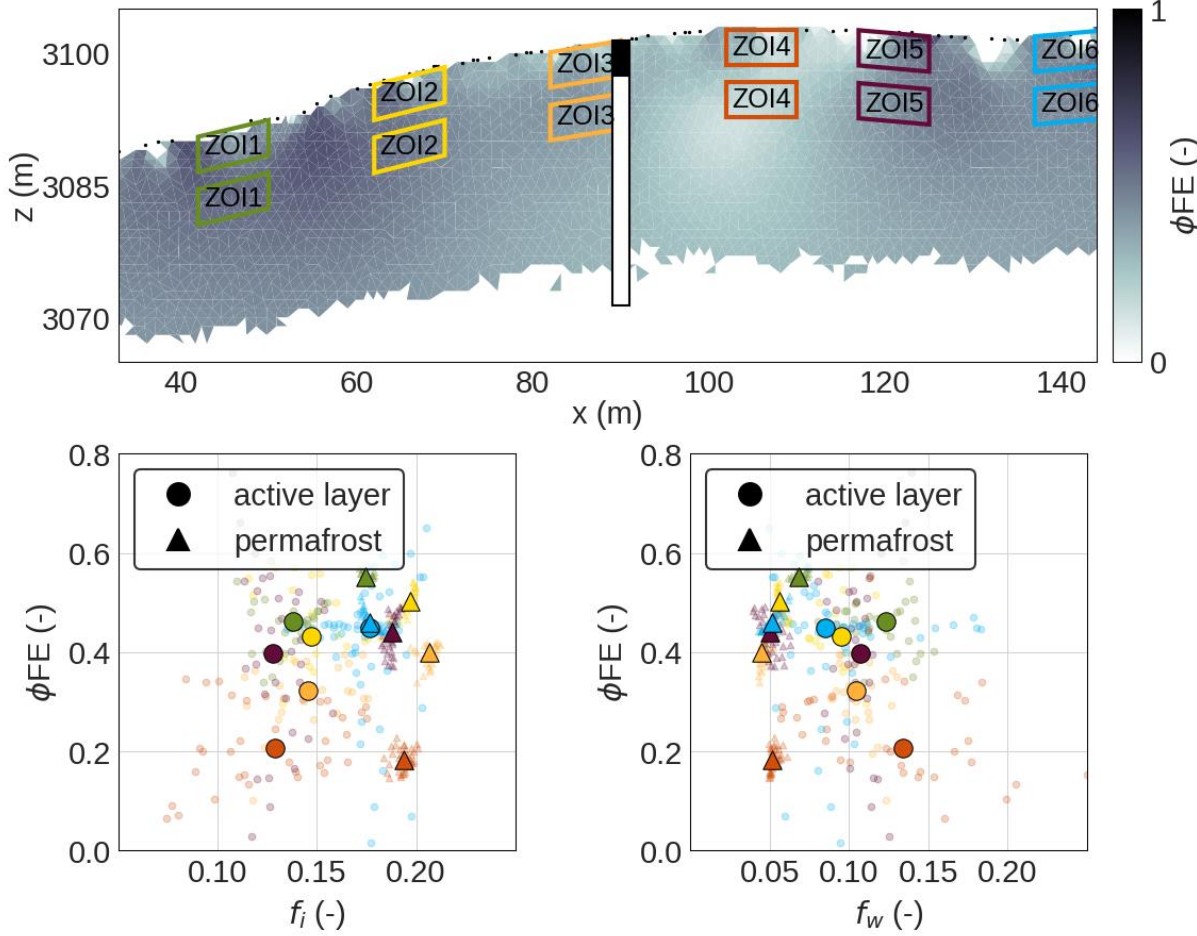

**Figure 13: Correlation between PJI-derived ice and water content estimates and the phase frequency effect for selected zones of interest (ZOIs) within the active layer and the permafrost body along the SIP monitoring profile.**

Additional investigations are needed to find a petrophysical model linking SIP parameters and ice content, where the $\phi FE$ could be a first option. Such a model could explicitly account for surface conduction and thereby help to reduce the uncertainty in the PJI inversion. However, to that end further SIP measurements at different sites and ice content are required (Maierhofer et al., 2021).

# 5 Conclusions and outlook

In this study, we presented time-lapse SIP imaging data collected at a mountain permafrost site during a measurement period of 3 years. The data were obtained in the frequency domain between 0.1-75 Hz using a combination of dipole dipole normal





and reciprocal and multiple gradient protocols. Our analysis highlights clear seasonal changes in the complex resistivity images
for all frequencies with an increase in late autumn and a decrease in early summer and most prominent changes at shallow
depths in the active layer. Characteristic changes in polarization can be attributed either to the freezing period in autumn
(September-November), before a permanent snow cover has been established, or to subsurface thawing processes in late spring
and early summer (June-July), when melting snow and rain lead to a decrease in phase values. During winter, complex
resistivity values increase to their maximum for all frequencies and depths but with a time lag and a smaller amplitude in
deeper zones, which is consistent with the delayed and damped changes in temperature, water content and ice content at greater
depths.

Complex resistivity-temperature relationships at the borehole location show a linear dependency for positive temperatures and
an exponential relation for subzero temperatures with different amplitudes for different substrates (debris and bedrock). Clear
hysteresis effects are observed in the freeze-thaw transition processes relating the change in polarization signals to a difference
in water content for these phase changes. Supercooling effects are observed through lower complex resistivity values during
freezing than during thawing, which is due to higher unfrozen water contents during freezing compared to thawing processes
at the same temperature. This interpretation is confirmed by laboratory measurements on rock samples from the site, which
upon freezing and thawing exhibit an absolute phase increase with decreasing temperature at higher frequencies, with the
general spectral behaviour being consistent with the known polarization properties of ice.

Our analysis based on field and laboratory freeze-thaw cycles as well as on spatial patterns of the SIP behaviour in freezing
conditions suggests that temperature, water and ice contents are dominant factors controlling the polarization. Considering that
field-SIP devices are often limited in the frequency range and might lack the possibility to recover the frequency maximum
(in our case the full process of ice relaxation) required to fit a dispersion model such as the Cole-Cole, this study focuses on
the investigation of the phase frequency effect, exploiting the spectral information gained through SIP field measurements in
a frequency range between 0.1-75 Hz. Yet, it is important that the frequency range in the SIP data extends above a sensitive
frequency, which corresponds to an inflection point between the low polarization values (due to the polarization of rock-water
interface) and the high values associated to the polarization of ice. We propose the resistivity phase frequency-effect ($\phi FE$) as
a proxy to discriminate between regions with no ice and regions richer in ice (i.e., also to discriminate temporal changes in ice
content). Laboratory and field data show a clear exponential increase of the temperature dependence of the $\phi FE$ during
freezing and higher amplitudes for higher ice contents (i.e., in the pores of the sediment sample and within the debris layer).
Annual changes of the $\phi FE$ at the field-scale are spatially heterogeneous, which is interpreted to be related to varying
surface/subsurface conditions, temperature changes as well as ice and water contents along the monitoring profile. Small
changes in temperature at depth can be resolved by SIP and result in a change of the phase frequency effect values.

When comparing the phase frequency effect with ice content estimates from collocated electric and seismic data sets using the
petrophysical joint inversion scheme (PJI), we observe a high consistency at the borehole location, but do not obtain the same
relationship for the entire imaging plane. These inconsistencies highlight the difficulty of obtaining ice content data sets for
validation, as also PJI-derived ice content estimates are based on various assumptions within the petrophysical model with





inherent uncertainties. Future investigations should focus on the integration of IP data in the PJI approach as structural constraints or as additional parameter in the underlying petrophysical model for an improved estimation of the ice content with applications in different permafrost landforms. In the presence of ice within the subsurface polarization increases and thus, spatially and temporally varying surface conduction has to be considered additionally to electrolytic conduction within the PJI to obtain reliable petrophysical parameters. Thus, further investigations are needed to find a petrophysical model linking IP and ice content such as a combination of the phase frequency effect with additional laboratory studies required to link the phase frequency effect with ice contents.

**Appendix A**

In Table A1, we present the measurement setup for SIP data collected along the monitoring profile between October 2019 and October 2022 and additional filtering parameters.

**Table A2: Filtering of SIP time lapse data – monitoring setup (electrode configuration and frequency range) and remaining quadrupoles after filtering for each measuring date**

| Measurement date | Electrode configuration | Frequency range (Hz) | Mean contact resistances ($\Omega$) | Mean injected current intensities (mA) | Remaining quadrupoles after filtering, 1/0.5 Hz | Remaining quadrupoles after filtering, 75 Hz |
|---|---|---|---|---|---|---|
| 2019-10-10 | DDsk3 N&R; MGsk0-3 | 0.5-225; 0.1-225 | 24180 | 10.4; 9.7 | 671; 612 | 366; 550 |
| 2019-11-12 | DDsk3 N&R; MGsk0-3 | 0.5-225; 0.25-225 | 22870 | 10.4; 9.5 | 670; 569 | 366; 473 |
| 2020-01-23 | DDsk3 N&R; MGsk0-3 | 0.5-225; 0.1-225 | 110140 | 2.3; 1.5 | 183; 339 | 123; 164 |
| 2020-02-15 | DDsk3 N&R; MGsk0-3 | 1-225; 0.1-225 | 116730 | 1.5; 1.1 | 141; 348 | 105; 207 |
| 2020-07-27 | MGsk0-3 | 0.5-225 | 24950 | 10.2 | 685 | 551 |
| 2020-07-29 | MGsk0-3 | 0.1-225 | 26810 | 9.2 | 605 | 441 |
| 2020-07-30 | DDsk3 N&R | 1-225 | 21920 | 10.6 | 699 | 386 |
| 2020-08-31 | MGsk0-3 | 0.1-225 | 26530 | 10 | 561 | 399 |
| 2020-09-19 | DDsk3 N&R; MGsk0-3 | 1, 7-225; 0.1-225 | 22320 | 9.6; 9.4 | 787; 618 | 309; 426 |



| 845 | 2020-10-27 | DDsk3 N&R; MGsk0-3 | 0.5-225; 0.1-225 | 64790 | 3.7; 10.9 | 437; 571 | 214; 357 |
| | 2020-11-26 | DDsk3 N&R; MGsk0-3 | 1-225; 0.1-225 | 120170 | 2.2; 3.6 | 263; 472 | 203; 313 |
| | 2021-01-21 | DDsk3 N&R; MGsk0-3 | 1-112.5; 0.1-225 | 156630 | 1.1; 1.4 | 90; 127 | 100; 139 |
| | 2021-02-25 | MGsk0-3 | 0.1-225 | 146150 | 0.4 | 218 | 232 |
| | 2021-04-01 | DDsk3 N&R; MGsk0-3 | 1-112.5; 0.1-225 | 138090 | 0.7; 0.5 | 121; 447 | 107; 348 |
| | 2021-05-04 | DDsk3 N&R; MGsk0-3 | 0.5-112.5; 0.1-225 | 136380 | 0.2; 0.6 | 154; 294 | 150; 246 |
| | 2021-06-04 | DDsk3 N&R; MGsk0-3 | 0.5-225; 0.1-225 | 88990 | 1.4; 1.4 | 318; 358 | 204; 301 |
| | 2021-07-01 | DDsk3 N&R; MGsk0-3 | 0.5-225; 0.1-225 | 38230 | 6.1; 5.2 | 469; 375 | 315; 265 |
| | 2021-08-03 | DDsk3 N&R; MGsk0-3 | 1-225; 0.1-225 | 28080 | 8.8; 8.7 | 656; 555 | 364; 379 |
| | 2021-08-09 | DDsk3 N&R | 1-225 | 38490 | 11.3 | 787 | 458 |
| | 2021-08-23 | DDsk3 N&R; MGsk0-3 | 0.1-225 | 32420 | 11.4; 9.8 | 588 | 421 |
| | 2021-09-30 | DDsk3 N&R; MGsk0-3 | 0.5-225; 0.1-225 | 35200 | 7.7; 8.5 | 751; 634 | 416; 476 |
| | 2021-11-05 | DDsk3 N&R; MGsk0-3 | 0.1-225 | 103160 | 1.9; 1.6 | 540 | 324 |



## Appendix B

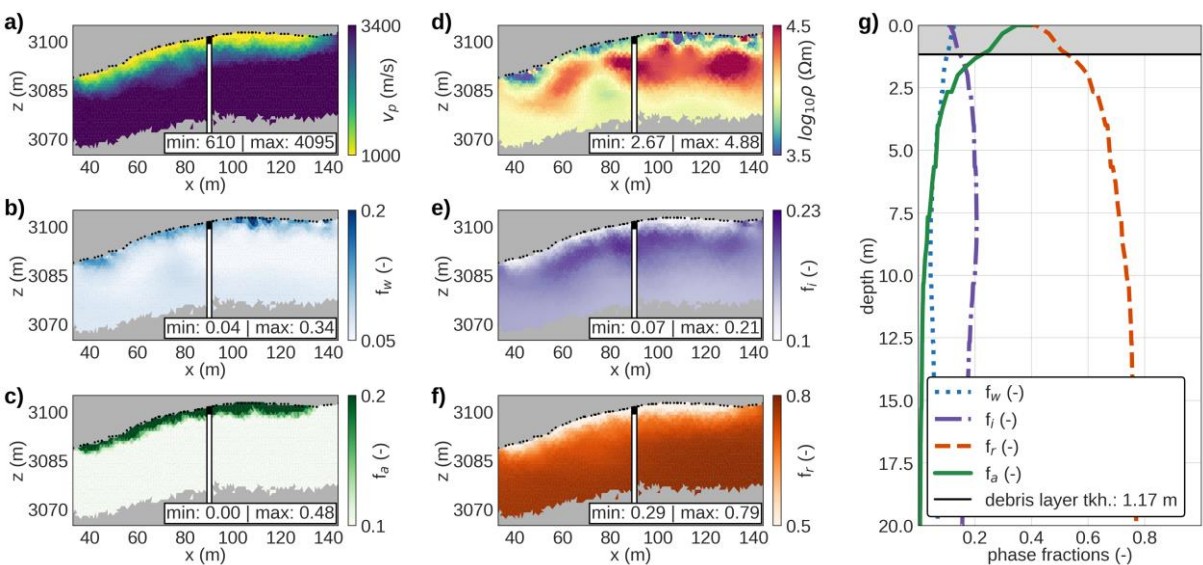

**Figure B1: Petrophysical joint inversion results for August 2020. (a-f) Imaging results in terms of geophysical parameters, i.e. the seismic P-wave velocity ($v_p$) and electrical resistivity ($\rho$), and petrophysical parameters, i.e the ice, water, air and rock contents resolved through the petrophysical joint inversion. (g) Virtual 1D logs extracted from the imaging results at the borehole location and estimated debris layer thickness.**

In Fig. B1 we show PJI results for seismic travel times and electrical resistivity measurements collected along the SIP monitoring profile. Differing velocities can be observed between the debris and bedrock with lowest velocities occurring in the near-surface within the porous debris layer ranging from roughly 500 to 1500 m/s, corresponding to a high air content. Maximum rock contents with a porosity of ~20% and minimum rock contents reaching a porosity of 70% are encountered in the bedrock and surface debris layer, respectively. Accordingly, the transition between debris and bedrock at the borehole location was estimated in a depth of 1.2 m by calculating the inflection point in rock content at the borehole position.

## Appendix C

The complex resistivity data were fitted for each frequency using a linear model for positive temperatures and an exponential model for subzero temperatures according to

$$\rho(T) = a \cdot T + b \tag{C1}$$

$$\rho(T) = a \cdot e^{(b \cdot T)} \tag{C2}$$

$$\phi(T) = a \cdot T + b \tag{C3}$$

$$\phi(T) = a \cdot e^{(b \cdot T)} \tag{C4}$$

with the coefficients for different frequencies for debris and bedrock summarized in table C1.



**Table C3: Coefficients for complex resistivity – temperature fits for two frequencies (0.5 and 75 Hz) and for two material compositions (debris and bedrock).**

| parameter | debris (0.5 Hz) | debris (75 Hz) | bedrock (0.5 Hz) | bedrock (75 Hz) |
|---|---|---|---|---|
| $\rho$ (T < 0°C) | $a = -0.0187$ | $a = -0.0185$ | $a = -0.0337$ | $a = -0.03$ |
| | $b = 3.9610$ | $b = 3.8982$ | $b = 4.034$ | $b = 3.9887$ |
| $\rho$ (T < 0°C) | $a = 4.0986$ | $a = -8.55e+03$ | $a = 4.0076$ | $a = -2.2639e+03$ |
| | $b = -0.0351$ | $b = -0.1589$ | $b = -0.0757$ | $b = -0.5124$ |
| $\phi$ (T > 0°C) | $a = -0.209$ | $a = -7.6766$ | $a = 0.0173$ | $a = -9.5084$ |
| | $b = 13.1938$ | $b = 70.7333$ | $b = 12.0308$ | $b = 83.1597$ |
| $\phi$ (T< 0°C) | $a = 32.8829$ | $a = -14.0334$ | $a = 8.9605$ | $a = 13.4545$ |
| | $b = 0.0109$ | $b = -176.9885$ | $b = 0.6501$ | $b = 8.1572$ |

*Data availability.* The borehole temperature data from the ARPA office, as well as the SIP field and laboratory data that support the findings of this study are available from the corresponding author upon request and will be provided through an online data repository after acceptance of the manuscript.

   *Author contributions.* TM, AFO and CHa designed the experimental setup, TM, ED, UMD and AFO planned and coordinated
the field logistics of the installation and conduction of the monitoring measurements and ED and UMD provided further background information of the site. TM, AFO and NR processed the geophysical field data and CM helped in the preparation of a figure. JL and AK collected and processed the laboratory data. AFO, CHa, CHi and TM interpreted the geophysical signatures and CHa, AFO, NR, CHi, AK, JL and TM discussed the results. TM led the preparation of the draft and wrote the major part of the text, where CHa, AFO and Chi helped essentially, while all authors contributed actively to the intermediate
and final version of the manuscript.

   *Acknowledgements.* This study is supported by the Swiss National Science Foundation (SNSF) and the German Research Foundation (DFG). We are furthermore thankful to the ARPA office for providing the borehole temperature data and ARPA as well as the cable car company Cervino S.p.A for the logistical support of our research activities at the Cervinia Cime Bianche
field site. We also thank Michel Isabellon, Lukas Aigner, Paolo Pogliotti, Doris Schlögelhofer, Alberto Carrera, Walter Loderer and Sarah Morard for the help in the preparation of the equipment, the installation of the geophysical monitoring profile and the collection of the geophysical monitoring data.

   *Competing interests.* The authors declare that they have no conflict of interest.




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
