# Peer review of "Spectral Induced Polarization imaging to monitor seasonal and annual dynamics of frozen ground at a mountain permafrost site in the Italian Alps"

_EGUsphere, 2023_

## Referee Comment (RC1)

Reviewer Recommendation and Comments for Manuscript Number egusphere-2023-671

**SPECTRAL INDUCED POLARIZATION IMAGING TO MONITOR SEASONAL AND ANNUAL DYNAMICS OF FROZEN GROUND AT A MOUNTAIN PERMAFROST SITE IN THE ITALIAN ALPS**

**Theresa Maierhofer et al.**

**Summary**

This manuscript presents a large spectral induced polarization (SIP) monitoring data set collected at an alpine permafrost site over the period of 3 years. Refraction seismic measurements in the field (late summer), broadband SIP measurements on sediment and rock samples at different temperatures (freezing and thawing cycles), and borehole temperature data complement the SIP monitoring data set and aid in the interpretation of the results.

SIP field and lab data show a characteristic increase of the resistivity phase shift at the high-frequency limit of the field data set around 100 Hz (higher frequencies are affected by electromagnetic coupling), which can be associated with the presence of water ice in the pore space. This interpretation is supported by the results of a joint petrophysical inversion approach of electrical and refraction seismic data, which resolves for the subsurface ice content (amongst other parameters), and by theoretical considerations (not discussed in detail in the manuscript). The authors propose the introduction of a quantity called the "phase frequency effect", which consists in the difference of the logarithms of the phase shifts observed at the highest and the lowest frequency, respectively, divided by the logarithms of these frequencies. This new quantity shows a good correlation with ice content estimates from the joint petrophysical inversion approach.

The manuscript is well written and structured. It provides all information needed to understand the collection, processing and interpretation of the presented data set. However, the text contains several instances of repetition and redundancies, which could be merged and streamlined to shorten the manuscript and improve its clarity. Furthermore, providing a more complete picture and discussion of the expected broadband response of ice-containing sediment and rock would improve the understanding of the effect underlying the newly introduced "phase frequency effect". Finally, the authors could provide some additional discussion on the risk of misinterpreting the effect of electromagnetic coupling in terms of ice content (both lead to increased phase values at high frequencies). These issues together with a number of specific comments and technical corrections listed below, should be straightforward to address. Therefore, I recommend accepting this interesting and relevant manuscript after a moderate revision.

**General comments**

Connection of phase frequency effect with ice relaxation at higher frequencies

As mentioned correctly by the authors, the main characteristic relaxation of ice-bearing sediment and rock is expected is most obvious in the kHz frequency range, where it often causes a clear peak in the phase spectrum. To allow all readers to understand how the increase at significantly lower frequencies (i.e., around 100 Hz) is related to this high-frequency peak, I suggest showing some broadband phase spectra from the lab measurements carried out within this project or a suitable theoretical model. Although this more complete picture would probably show that the suggested approach is not yet the best of the (theoretically) possible methods for

ice-content estimation, the contribution is very valuable as it shows a suitable and well-justified workaround as long as no multi-channel high-frequency SIP devices are available.

Influence of inductive/capacitive coupling at high frequencies

An increase of phase values at high frequencies does not necessarily indicate a high ice content. More commonly, high phase values at high frequencies are related to inductive and/or capacitive effects. I suggest including a more rigorous discussion of the risk of misinterpreting such coupling effects in terms of high ice content. It should also be mentioned that such a misinterpretation can best be reduced by improving measurement capabilities at high frequencies (100 kHz and higher).

Potential of shortening the manuscript

The manuscript could be shortened significantly by systematically removing redundancies (data presented repeatedly in various figures and repetitions of explanation and concepts in the text). For (only some) examples that might help to start streamlining the manuscript, please see specific comments below.

**Specific comments and technical corrections**

Line 3: frequency-dependence -> frequency dependence (without hyphen) (orthography)

Line 30: physical properties -> more specifically: thermal state (or similar)

Line 45: parts -> part (grammar)

Line 46: is transformed into ice -> freezes

Line 56: Add "which has not yet been implemented in the PJI."

Line 59ff: "… while the imaginary component relates to surface conductivity arising from the accumulation and polarization of charges at the EDL formed in the ice-water

or rock-water interface as well as protonic defects in ice surface." Without further discussion, this list of effects is not complete. I recommend adding an "e.g." and defining the frequency range of interest (too late for this statement, if introduced in the next sentence).

Line 61ff: Please note that the study of Mudler clearly exceeds the usual IP frequency range by at least two orders of magnitude. As stated in later in the discussion, ice-related relaxation effects most clearly show up at frequencies >1000 Hz. Already at this point, the reference to the study by Mudler et al. (2022) should be accompanied by a more detailed discussion of the differences of "usual" SIP measurements and high-frequency IP (HFIP) measurements (see my general comment).

Line 72: "retrieved ice volumes" -> "were able to estimate ice volumes at a … from electrical resistivity data (or similar)".

Line 73: What is a talus slope? Please add a brief description.

Line 89: "measured surface conductivity" -> How can the surface conductivity be measured directly? Maybe better "the effect of surface conductivity"?

Figure 1: Please add detailed information on sources of the background images: Digital elevation model in panel 1, satellite data in panels 2 and 3. In the caption, mention that the satellite background picture is overlain by an orthophoto of the survey area including contour lines (derived from the orthophotos?). What do the blue shadings in panel 2 mean? What does the single blue circle south of the label "MON" indicate?

Line 134: "monitoring period" -> "SIP monitoring period" (as other monitoring periods of the site started as early as 2004...)

Line 137: according to Fig. 2 b and c, temperatures in DBH during summer 2022 were even higher. Please check!

Figure 2: In sight of the total length of the manuscript, I recommend reducing the data shown in this figure: Are precipitation and air/surface temperature strictly necessary for the discussion of the geophysical data or could they be shown in the supplementary material? Ground volumetric water content is redundant with the data shown in Fig. 11. Panels b and c show the same data in two different ways. Part of the same temperature data is shown again in Fig. 12.

Line 169, equation (1): "-" -> "+", otherwise $\rho''$ needs to be defined as the negative of the imaginary part of the resistivity.

Line 170, equation (3): Remove "-" (see above)

Line 186ff: Should read "...between the logarithms of the low-frequency and high-frequency... normalized by the difference of the logarithms of..." or similar.

Line 208: Add "remaining" or similar before "liquid pore water phase"

Line 229: Add "(in percent)" following "volumetric water content" to motivate the multiplication by 100.

Line 233 / equation (9): Shouldn't the equation rather read $S_W = e^{\frac{\ln\left(\frac{\rho_w \phi^{-m}}{\rho_{bcorr}}\right)}{n}}$? I also suggest expressing the equation in a simpler form eviting the exponential, the logarithm and the negative sign of the cementation exponent as follows: $S_W = \sqrt[n]{\frac{\rho_w}{\rho_{corr}\phi^m}}$.

Line 236: Instead of "represents a temperature correction applied to the bulk electrical resistivity data", consider writing "is the temperature-corrected bulk electrical resistivity"

Line 237, equation (10): Please consider introducing this temperature correction together with the temperature correction in Line 200, equation (6).

Line 262f: "using 4 potential dipoles (with lengths of...)" -> "using 4 potential dipole lengths of ..."

Line 268: Do these current injections refer to the RS-check or the actual measurement. Please specify in this sentence.

Line 269: "acquired"-> "observed"

Line 275: "Corona pandemic" -> "COVID-19 pandemic"

Line 299f: Please justify/motivate the use of equal error parameters for all frequencies in more detail. Usually, high-frequency data is more affected by electromagnetic coupling than low-frequency data. Why would a frequency-independent error model still be a good assumption?

Line 307ff: Please mention that the computation of the RMS includes a normalization of the individual misfits by the error taken from the error model. Otherwise, the dimensionless target value of 1 does not make sense.

Line 315, equation (14): Please provide a more information on the determination of R(x,z). E.g. are $m_1$ and $m_2$ fixed as suggested by the definition in the next line or do they vary with x and z as suggested by equation (14)? What are the values of $m_{1r}$ and $m_{2r}$?

Line 316f: Are values of $R$ computed for resistivity and phase images and applied separately to the corresponding images? Or do you use one "mask" for all images?

Line 318: Is the DOI index equal to R? In this case, could you either use DOI OR R?

Line 320, Figure 3: Here, the computation of the DOI analysis appears before the inversion step. From the explanations above, I understood that this analysis was based on the inversion output ("mean… values observed in all SIP images"). Please check and – if necessary – explain in more detail.

Line 345f: "same values" -> "same values of porosity and seismic velocities"

Line 349ff: Obviously, the selection of a constant value of the surface conductivity is empirically useful. Could this selection also be physically sensible/meaningful? Please add a short discussion of the implications of this assumption.

Line 358ff: How was the saturating fluid prepared (tap water with NaCl)? Please provide some more detail.

Line 268: Actually, this section is rather a "Results and discussion" section, as most of the results are already discussed in some detail. In particular, in this section the text could be shortened by streamlining (in German "straffen") the presentation of results. E.g., lab and field data could be discussed together instead of separately, which leads to many redundancies in the discussion.

Line 374: "inflection point in electrical resistivity": The depth slices in Figure 4 are not suitable to show an inflection point in the depth profile. Please consider discussing this concept based on depth profile (e.g. at the positions of the two boreholes).

Line 376: In the Figure, all resistivity values are shown as log10-scaled values, while the text refers to unscaled values. This makes it unnecessarily hard to follow the discussion. Please consider replacing the corresponding labels of the colour bar in the Figure by unscaled values (e.g. 3.75 -> 5620 Ωm) or similar to make the information more easily accessible.

Line 382 / caption to Figure 4: Please add meaning of white contour lines (surface elevation).

Line 387: "The baseline SIP imaging results…" please consider adding "along the permanent monitoring profile"

Line 410 / Figure 5:

- Please consider numbering photos in (a) from left to right 1-2-3. It is not clear, why the figures are numbered in inverted order.
- Imaginary resistivity and phase values of 0.5 Hz and 75 should be plotted on the same scale (colour bar limits) to facilitate a direct comparison of these pairs of sections (as discussed in the main text).

Line 419: "including warm (…) and phase (…)" consider adding "in the active layer" (this information is given in the Figure´s caption but would help here in the main text).

Line 423: "with no" -> "with only small"

Line 416: "above" -> "starting at"

Line 449 / Figure 6:

- Can the field data in panel a) be presented in the same fashion as the lab data in panels b)? I.e. could the curves be coloured based on the temperatures measured in the boreholes to facilitate the comparison of field and lab data even more?

- For understandable reasons (comparison with field data), lab data is only visualized up to 100 Hz although it has been measured up to 45 kHz. Please consider providing the complete spectral information in the supplementary material.

Line 459: Please provide more information on why and at which frequency such a peak would be expected (e.g. "…which according to… would be expected around … Hz…")

Line 481 / Figure 7: Could the curves for freezing and thawing be combined in one plot? This would make it easier to appreciate the effect of hysteresis between the two directions of temperature change.

Line 482 / caption of Figure 7: Please provide information on the measurement frequency displayed here.

Line 505 / Figure 8: Is it a fortunate coincidence that the ice-content data and the phase frequency effect data at the borehole position correlate that well (see Figure 13)? Why show the good correlation of the two sub-datasets around the borehole here and discuss the limitations separately later in the manuscript? Neither the ice-content nor the phase frequency effect data is limited to the location around the borehole. Regarding the streamlining (shortening) of the text, please consider discussing this correlation only once.

Line 554: Discussing Figure 10c, the authors state "… we also observe at the field-scale lower $\rho'$, $\phi$ and $\phi FE$ during freezing (from October to January) than during thawing (from May to July)…" Figure 10c does not really support this statement. At any given temperature around 0 °C, I observe more low values in warm colours (summer months = thawing) than low values in cold colours (winter months = freezing). Please check!

Line 556-559: Redundant: Has already been discussed above.

Line 577-595: Please consider discussing possible sources and effects of data error in a separate subsection (i.e. with a separate title).

Line 596: Delete "so-called"

Line 604-606: Redundant.

Line 626: Please define "R".

Line 641 / Figure 11: Please check equation in panel b) (see comment on equation (9)).

Line 701 / Figure 12c): Please consider showing the temperature data only down to 15 m depth to facilitate the comparison with the phase frequency effect data in Figure 12d).

Line 715: It would be helpful for the understanding, if some suitable lab data or modelled data containing the "ice peak" in the frequency range between 1 and 45 kHz was shown earlier in the manuscript (best where it is mentioned/discussed first).

Line 739: "Blacheck" -> "Blaschek"

Line 870: Data availability statement: A link to the mentioned repository should be provided during the review process to allow the reviewers to assess the information provided.

03/07/2023, Matthias Bücker
TU Braunschweig
Institute for Geophysics and extraterrestrial Physics
Braunschweig, Germany

---

## Referee Comment (RC2)

**Summary**

The manuscript covers and introduces a new insight into application of the SIP method for cryological research. It shows that the SIP method is sensitive to the temperature, and therefore, temporal changes of the surface, and confirms that monitoring datasets provide valuable information about permafrost evolution.

The Introduction, Material and Methods, Discussion and Conclusion are well written. However, in its current format the manuscript is too lengthy and at times the overarching message is lost in the details. It is advisable to attempt to make the Results section further simplified / concise to improve readability, specifically sections 3.2 – 3.5. It would be helpful to highlight clear messages, so the reader is satisfied to have understood the section before moving onto the next.

Another important aspect is to highlight the deficiencies / drawbacks of the SIP field measurement system applied here. The system measures up to 225 Hz but (I am assuming) due to coupling effects and higher errors at higher frequencies, data only up to 75 Hz was considered. So, there is a clear need for newer or improved measurement systems that are stable at higher frequencies, specifically for permafrost research dealing with polarisation of ice – which is a topical research area.

An overall comment for the figures in the manuscript is that efforts should be made to reduce text crowding and / or remove panels that do not add value to the figure. Consistent colour scales / themes should be used.

The results presented here are interesting and exciting, and adds further knowledge to this emerging research field, thank you for sharing this. Please see below for detailed comments on text and figures.

**Line-by-line comments:**

Abstract

1. Line 12: Consider reformulation: We investigate the application of spectral induced polarization (SIP) monitoring, to understand seasonal and annual variations of freeze-thaw processes in permafrost, by examining the frequency dependence of subsurface electrical properties. (grammar)
2. Line 15: The SIP imaging results were interpreted in conjunction with complementary seismic and borehole datasets. (grammar).
3. Line 24: Comma after "observed in field data". (grammar)

4. Line 28: Comma after (Pepin et. al., 2015) (grammar)

5. Line 31: Consider simplifying to "and water storage capacities" (grammar)
6. Line 62: Mudler et. al. 2022 discuss that polarisation effects go beyond 1000 Hz.

7. Line 70: Change to "broadband". (grammar)
8. Line 75: Considering changing "but different IP responses." to "but they can be distinguished due to their distinct IP responses." (grammar)

9.  Line 87: Aren't points 1 and 2 giving the same message/circular loop? Surface conductivities can be determined through SIP to improve ice-content estimation using petrophysical models. The sentence structure is slightly confusing.
10. Line 92: What does signatures mean here?
11. Line 94: Comma after "state" (grammar)
12. Line 95: Consider adding the abbreviation ($\phi$FE) as it is being introduced in the main article for the first time.

13. Line 98: Unsure about the difference between volumetric water content and unfrozen water content. Can you explain further?
14. Line 99: Perhaps change "sustain" to "evaluate". (grammar)
15. Line 107: Consider changing "Consist of" to "consisting of". (grammar)
16. Line 109: Perhaps a quick explanation of Gelifluction?

17. Figure 1: The abbreviation "MON" in the bottom image is not explained? I believe this is the ERT/RST profile. Consider indicating this in the caption.
    There is a blue dot below "MON" is this intentional?
    Legend says meteorological sensors (plural) but caption says sensor (singular)
    Change "SIPM the SIP monitoring profile" to "the SIP monitoring profile (SIPM).
    Overall, it would be good to be consistent that the IDs (e.g., MON, MD, DBH, etc) are indicated in the caption. It helps when jumping back and forth.

18. Section 2.1 describing the field site can be shortened by removing information repeated further down manuscript.

19. Figure 2: I'm not sure if panel (c) is necessary here? It is difficult to follow what the image is communicating. Panel (b) is sufficient in showing the depth at which the temperature is 0 degrees Celsius.

20. Line 172: Can you explain why the range 0.1-1000Hz? What does commonly imply?

21. Line 225: Does the symbol $\phi$ here represent porosity? In this case perhaps it's better to represent it with capital $\Phi$ to avoid confusion with phase shift. Also indicate that this term stands for porosity in text.

22. Line 223: What is $P_{corr}$ here? This has been explained in line 236. Perhaps it can be brought one sentence forward?
23. I'm assuming Pore water resistivity is $p_w$. The abbreviation should be added in brackets.
24. Line 240: Should the term by $S^0_w$ instead? To keep it consistent with the terminology of $P_f$ and $P_o$

25. Line 275: Consider changing corona pandemic to "COVID-19 Pandemic". (grammar)

26. Line 315: What term does "R" denote?

27. Consider combining section 2.4 and 2.5 as Complementary geophysical datasets to shorten length of the manuscript.

28. Line 389: Can you explain why the 75 Hz frequency is used at the high frequency, even though measurements were conducted up to 225 Hz?
29. Line 392: Can you recheck the sentence structure for "In general comparatively… "
Please recheck the terms and consider simplification. (grammar)

30. Line 406: In general, the information contained in these long brackets are hard to follow and hinder the flow of reading. Perhaps consider denoting the graph with symbols and providing this parameter breakdown in a table.

31. Consider making section 3.2 more concise.

32. Line 444: Can you explain why the field data shows higher phase angles? I understand that it corresponds to similar surface cover of small amphibolite rocks. So, it should be similar values as that in the lab? This is an important point to discuss and also include in conclusion – does this have further consequences? Is it a drawback of the measurement device or inversion?

33. Figure 8 d-e and f-g show the same dataset in different ways. Perhaps only show panels d-e and indicate correlation r2 values in text.

34. Line 546: The highlighted sentences can be condensed such that "the results correspond with those from laboratory, indicating ɸFE is higher for loose sediment at subfreezing temperatures, and, a lower freezing/melting is observed for the bedrock." The length of the manuscript now starts to feel cumbersome so care should be taken to reduce repetition of information where possible.
35. Line 554: Should it be "higher" ρ', ɸ and ɸFE during freezing?

36. Figure 10c: It is recommended to amend figure 10, specifically the location of 10(c) as it's confusing where these measurements are conducted – and how is it linked to 10(a) and 10

(b). For starters you can indicate that measurements were conducted in the debris layer in the caption or somewhere else in the panel itself. Secondly, move the panel away from a. and b. panels to make it obvious to the reader that it's a different type of result that we are viewing.

37. Line 597: Consider changing to "use of Phase Frequency Effect ($\phi$FE) as a proxy…" (grammar)

38. Line 615: Can you further explain why the $\phi$FE is increasing with increasing temperatures?

39. Figure 11: General comment that this figure comes across very crowded. Consider reducing x axis labels and utilizing major / minor grid lines. Secondly, perhaps the text from can be moved to outside / adjacent to the plot? The colour scale should also be consistent and easily to understand for dry and wet conditions.
Lastly, the colour scale of the resistivity vs temperature plot should also be consistent throughout. Also, this panel is not explained in the caption or in text. Is there any impact by removing this panel altogether?

40. Line 643: Should it say various depths (0.6 – 20 m)?

41. Line 761: Please recheck if AL abbreviation has been introduced before

42. Line 816: Indicate that these "are the dominant factors controlling the polarisation at the frequencies measured in this study."

[revised manuscript text omitted]

---

## Author Response (AR1)

Reviewer Recommendation and Comments for Manuscript Number egusphere-2023-671

**SPECTRAL INDUCED POLARIZATION IMAGING TO MONITOR SEASONAL AND ANNUAL DYNAMICS OF FROZEN GROUND AT A MOUNTAIN PERMAFROST SITE IN THE ITALIAN ALPS**

**Theresa Maierhofer et al.**

**Summary**

**This manuscript presents a large spectral induced polarization (SIP) monitoring data set collected at an alpine permafrost site over the period of 3 years. Refraction seismic measurements in the field (late summer), broadband SIP measurements on sediment and rock samples at different temperatures (freezing and thawing cycles), and borehole temperature data complement the SIP monitoring data set and aid in the interpretation of the results.**

**SIP field and lab data show a characteristic increase of the resistivity phase shift at the high-frequency limit of the field data set around 100 Hz (higher frequencies are affected by electromagnetic coupling), which can be associated with the presence of water ice in the pore space. This interpretation is supported by the results of a joint petrophysical inversion approach of electrical and refraction seismic data, which resolves for the subsurface ice content (amongst other parameters), and by theoretical considerations (not discussed in detail in the manuscript). The authors propose the introduction of a quantity called the "phase frequency effect", which consists in the difference of the logarithms of the phase shifts observed at the highest and the lowest frequency, respectively, divided by the logarithms of these frequencies. This new quantity shows a good correlation with ice content estimates from the joint petrophysical inversion approach.**

**The manuscript is well written and structured. It provides all information needed to understand the collection, processing and interpretation of the presented data set. However, the text contains several instances of repetition and redundancies, which could be merged and streamlined to shorten the manuscript and improve its clarity. Furthermore, providing a more complete picture and discussion of the expected broadband response of ice-containing sediment and rock would improve the understanding of the effect underlying the newly introduced "phase frequency effect". Finally, the authors could provide some additional discussion on the risk of misinterpreting the effect of electromagnetic coupling in terms of ice content (both lead to increased phase values at high frequencies). These issues together with a number of specific comments and technical corrections listed below, should be straightforward to address. Therefore, I recommend accepting this interesting and relevant manuscript after a moderate revision.**

Thank you very much for the constructive and helpful comments, which we appreciate a lot! We considered your suggestions carefully and address them in the following in detail.

We admit that our Results Section in the original manuscript is too long and important details get lost. We tried to simplify the entire manuscript, especially the Results Section by merging and removing repetitions and improving its readability. Therefore, we summarized field and laboratory results instead of describing each result separately as in the case of Section 3.2. We made sections 3.2-3.5 more concise and highlight now clear messages. Additionally, we tried to reduce subplots, text and panels from different figures that did not add value to the figure (for example Fig. 2, 7, 8 and 11).

We definitely agree with your comment concerning the need for a more complete picture of the expected broadband response of ice-containing sediment and rock and therefore added an additional Figure of the broadband response of SIP laboratory measurements on a solid rock sample and a loose

sediment sample for a frequency range between 10 mHz and 45 kHz during controlled freeze-thaw cycles (+20°C to -40°C to +20°C) to the Appendix B.

We acknowledge your advice of a further discussion on the risk of misinterpreting the effect of electromagnetic coupling in terms of ice content. We therefore added a subsection "Reliability of SIP monitoring measurements" within the Discussion Section of the revised version of the manuscript, where we quantitatively assess the effect of a high sample resistance and electromagnetic fields (i.e., parasitic capacitive coupling (PCC)) on SIP measurements and examine the accuracy of the phase measurements collected with the DAS-1 measurement device that we used within our SIP monitoring measurements. We built an electric circuit using commercial resistors and capacitors with known values (see new Figure 11 in the revised version of the manuscript) as suggested by different authors (e.g., Revil and Skold, 2011; Wang and Slater, 2019) and performed impedance measurements with the DAS-1 field device and the Portable Spectral Induced Polarization (PSIP) laboratory unit (Ontash & Ermac Inc., NJ). Such analysis shows that phase errors due to PCC of the instrumentation or cables can be neglected for frequencies between 0.1 and 75 Hz with the phase frequency effect being a stable measure. At frequencies above 100 Hz, phase values strongly deviate from the theoretical response, highlighting the need for correction methods such as presented by Wang and Slater (2019) when modelling and interpreting phase measurements at higher frequencies. If sample resistance exceeds $1\,k\Omega$, as could be the case for winter measurements, phase errors increase leading to a high data uncertainty. Hence, our analysis focuses on the summer period as well as the freezing and thawing period. For a detailed interpretation of winter SIP measurements, future studies should consider the simulation of the effect of high contact impedances of current and potential electrodes, which additionally increases phase errors as pointed out by Ingemann-Nielsen (2006).
Such studies show that an increase in phase values with increasing frequency observed within our study does not automatically indicate a high ice content. By this discussion, we believe that we accordingly addressed your comment.

We hope that our adaptions answer your comments and we thank you again for your helpful suggestions.

**General comments**
**Connection of phase frequency effect with ice relaxation at higher frequencies**
**As mentioned correctly by the authors, the main characteristic relaxation of ice-bearing sediment and rock is expected is most obvious in the kHz frequency range, where it often causes a clear peak in the phase spectrum. To allow all readers to understand how the increase at significantly lower frequencies (i.e., around 100 Hz) is related to this high-frequency peak, I suggest showing some broadband phase spectra from the lab measurements carried out within this project or a suitable theoretical model. Although this more complete picture would probably show that the suggested approach is not yet the best of the (theoretically) possible methods for**

**ice-content estimation, the contribution is very valuable as it shows a suitable and well-justified workaround as long as no multi-channel high-frequency SIP devices are available.**

We agree with the comment of the reviewer and added some broadband phase spectra from the laboratory in the supplementary material in order to show how the increase at lower frequencies is related to the high-frequency maximum in the phase spectrum. Laboratory data is less affected by sources of noise limiting field measurements and thus, we discuss the polarization of ice based on the laboratory response. However, our lab data does not reveal a clear peak at high frequencies, which could be due to different polarization mechanisms at high frequencies such as Maxwell Wagner polarization. However, the discussion of the high-frequency laboratory data is beyond the scope of this manuscript and will be presented in another manuscript focusing on the laboratory measurements. Nonetheless, the proposed $\phi FE$ is able to capture the thermal dynamics we want to solve from SIP data (see Figure 7).

**Influence of inductive/capacitive coupling at high frequencies**
**An increase of phase values at high frequencies does not necessarily indicate a high ice content. More commonly, high phase values at high frequencies are related to inductive and/or capacitive effects. I suggest including a more rigorous discussion of the risk of misinterpreting such coupling effects in terms of high ice content. It should also be mentioned that such a misinterpretation can best be reduced by improving measurement capabilities at high frequencies (100 kHz and higher).**

Thank you very much for this suggestion, we added a new subsection 4.1 "Reliability of SIP monitoring measurements" within the Discussion section, where we discuss the risk of misinterpreting electromagnetic coupling effects in terms of ice content.
We therefore added a new paragraph in lines 649-693, where we discuss potential distortions in the induced polarization data due to electromagnetic fields observed from different authors (Binley et al. 2005; Flores Orozco et al., 2018; Ingeman-Nielsen, 2006; Kemna et al. 2012; Wang and Slater, 2019; Zimmerman et al., 2008) leading to significant phase errors at higher frequencies.
Within electric circuit tests in the laboratory (new Fig. 11) we examine the accuracy of the phase measurements collected with the field measurement device and observe high accuracy for both magnitude and phase measurements with phase errors less than 0.5 per cent at low frequencies and phase frequency errors smaller than 4 per cent between 0.1 and 75 Hz if resistances stay below 1 kHz. At frequencies above 100 Hz, phase values strongly deviate from the theoretical response, highlighting the need for correction methods such as presented by Wang and Slater (2019) when modelling and interpreting phase measurements at higher frequencies.

**Potential of shortening the manuscript**
**The manuscript could be shortened significantly by systematically removing redundancies (data presented repeatedly in various figures and repetitions of explanation and concepts in the text). For (only some) examples that might help to start streamlining the manuscript, please see specific comments below.**

We tried to shorten the manuscript by reducing the number of subplots in Figures 2, 7, 8 and 11 and streamlining the Results section. We added a subsection within the Discussion section as it was requested from both reviewers and tried to compensate by removing redundancies in the other sections.

**Specific comments and technical corrections**
Thank you for the constructive and helpful comments. We considered each comment carefully and address them in detail in the following text.

1) **Line 3: frequency-dependence -> frequency dependence (without hyphen) (orthography)**

We adapted the entire manuscript accordingly by removing the hyphen.

2) **Line 30: physical properties -> more specifically: thermal state (or similar)**

We agree with the reviewer and changed it to thermal state.

3) **Line 45: parts -> part (grammar)**

Thank you for the indication, we corrected it accordingly.

4) **Line 46: is transformed into ice -> freezes**

We did not change the phrase, since the phase change from liquid water to ice and the term freezing are not equal, as porewater with a high salinity can be already frozen while there is still no ice within the pore-space.

5) **Line 56: Add "which has not yet been implemented in the PJI."**

We added the phrase accordingly.

6) **Line 59ff: "… while the imaginary component relates to surface conductivity arising from the accumulation and polarization of charges at the EDL formed in the ice-water or rock-water interface as well as protonic defects in ice surface." Without further discussion, this list of effects is not complete. I recommend adding an "e.g." and defining the frequency range of interest (too late for this statement, if introduced in the next sentence).**

Thank you very much for this comment, we agree with this point and followed the recommendation of the reviewer.

7) **Line 61ff: Please note that the study of Mudler clearly exceeds the usual IP frequency range by at least two orders of magnitude. As stated in later in the discussion, ice-related relaxation effects most clearly show up at frequencies >1000 Hz. Already at this point, the reference to the study by Mudler et al. (2022) should be accompanied by a more detailed discussion of the differences of "usual" SIP measurements and high-frequency IP (HFIP) measurements (see my general comment).**

We added a description in lines 67-70 focusing on the difference between SIP measurements commonly collected between 0.1 to 1000 Hz and high-frequency IP measurements, which are required to capture the full ice relaxation process.

8) **Line 72: "retrieved ice volumes" -> "were able to estimate ice volumes at a … from electrical resistivity data (or similar)".**

Thank you for the suggestion we changed the phrasing.

9) **Line 73: What is a talus slope? Please add a brief description.**

We added a brief description about this typical permafrost landform.

10) **Line 89: "measured surface conductivity" -> How can the surface conductivity be measured directly? Maybe better "the effect of surface conductivity"?**

We agree with the reviewer and adapted the sentence accordingly.

11) **Figure 1: Please add detailed information on sources of the background images: Digital elevation model in panel 1, satellite data in panels 2 and 3. In the caption, mention that the satellite background picture is overlain by an orthophoto of the survey area including contour lines (derived from the orthophotos?). What do the blue shadings in panel 2 mean? What does the single blue circle south of the label "MON" indicate?**

Thank you very much for raising these points. We added the information on the background images to Figure 1 and to the caption of Figure 1. The blue circle in panel 3 represents the position of the base station of the DGPS. We deleted the point and also the blue shadings in panel 2.

12) **Line 134: "monitoring period" -> "SIP monitoring period" (as other monitoring periods of the site started as early as 2004…)**

We agree with the reviewer that this is misleading and corrected it within the entire manuscript.

13) **Line 137: according to Fig. 2 b and c, temperatures in DBH during summer 2022 were even higher. Please check!**

We looked into the data and indeed, the temperatures varied between -0.3 and -1 °C during the SIP monitoring period in the deep borehole in a depth of 10 m. In our version of the manuscript, we rounded the values, but corrected it in the new version.

14) **Figure 2: In sight of the total length of the manuscript, I recommend reducing the data shown in this figure: Are precipitation and air/surface temperature strictly necessary for the discussion of the geophysical data or could they be shown in the supplementary material? Ground volumetric water content is redundant with the data shown in Fig. 11. Panels b and c show the same data in two different ways. Part of the same temperature data is shown again in Fig. 12.**
We agree and drastically reduced the data shown in Figure 2.

15) **Line 169, equation (1): "-" -> "+", otherwise $\rho''$ needs to be defined as the negative of the imaginary part of the resistivity.**
We changed the sign in Equation 1.

16) **Line 170, equation (3): Remove "-" (see above)**
Due to the sign-change in Equation 1, we also removed it in Equation 3.

17) **Line 186ff: Should read "…between the logarithms of the low-frequency and high-frequency… normalized by the difference of the logarithms of…" or similar.**
Thank you for the suggestion, we adapted lines 205-206 accordingly.

18) **Line 208: Add "remaining" or similar before "liquid pore water phase"**
We modified the sentence accordingly.

19) **Line 229: Add "(in percent)" following "volumetric water content" to motivate the multiplication by 100.**

20) We modified the sentence accordingly.

21) **Line 233 / equation (9): Shouldn't the equation rather read $Sw = e_{\ln(\rho w \phi - m \rho bcorr)n}$? I also suggest expressing the equation in a simpler form eviting the exponential, the logarithm and the negative sign of the cementation exponent as follows: $Sw = \sqrt[mn]{\rho w \rho corr \phi}$.**
Thank you for the indication, we expressed the equation in a simpler form.

22) **Line 236: Instead of "represents a temperature correction applied to the bulk electrical resistivity data", consider writing "is the temperature-corrected bulk electrical resistivity"**
We rephrased the sentence accordingly.

23) **Line 237, equation (10): Please consider introducing this temperature correction together with the temperature correction in Line 200, equation (6).**
We agree and shifted equation 10 to section 2.3.2.

24) **Line 262f: "using 4 potential dipoles (with lengths of…)" -> "using 4 potential dipole lengths of …"**
We adapted the sentence as suggested by the reviewer.

25) **Line 268: Do these current injections refer to the RS-check or the actual measurement. Please specify in this sentence.**
The current injections refer to the actual measurement. We agree that mentioning the current densities within the description of contact resistances is misleading and moved it to lines 270-272 within the general description of data acquisition.

26) **Line 269: "acquired"-> "observed"**
Thank you for the suggestion, we adapted the sentence accordingly.

27) **Line 275: "Corona pandemic" -> "COVID-19 pandemic"**
Thank you for the suggestion, we adapted the phrase accordingly.

28) **Line 299f: Please justify/motivate the use of equal error parameters for all frequencies in more detail. Usually, high-frequency data is more affected by electromagnetic coupling than low-frequency data. Why would a frequency-independent error model still be a good assumption?**
As mentioned within the Discussion Section, we used different error parameters for each frequency considering the frequency dependence observed in our data. The inversion with different error parameters for each data set collected at different frequencies and times was adapted in the final steps of the preparation of our manuscript, which we forgot to change also in the Material and Methods Section. Thank you very much for spotting this error, we clarified our approach also in the Material and Methods Section in lines 306-308 and added a justification/motivation as follows: Similar to the study of Flores Orozco et al. (2011), we used

different error parameters for the inversion of spectral data to consider the frequency dependence observed in our data with higher misfits detected for higher frequencies compared to lower frequencies.

**29) Line 307ff: Please mention that the computation of the RMS includes a normalization of the individual misfits by the error taken from the error model. Otherwise, the dimensionless target value of 1 does not make sense.**

Thank you for the indication, we corrected lines 336-339 accordingly.

**30) Line 315, equation (14): Please provide a more information on the determination of R(x,z). E.g. are $m_1$ and $m_2$ fixed as suggested by the definition in the next line or do they vary with x and z as suggested by equation (14)? What are the values of $m_{1r}$ and $m_{2r}$?**

Thank you for spotting this error, we exchanged the variables $m_1$ and $m_{1r}$ in the text. The variables $m_1$ and $m_2$ are the model vectors of complex conductivities of the individual cells of the underlying finite-element mesh and vary with x and z. We calculated resistivity and phase images for all frequencies with constant reference models $m_{1r}$ and $m_{2r}$. We clarified the misunderstanding within this paragraph.

**31) Line 316f: Are values of $R$ computed for resistivity and phase images and applied separately to the corresponding images? Or do you use one "mask" for all images?**

Thank you for the indication, to specify this detail in our manuscript, we added a small paragraph to the Material and Methods section.

**32) Line 318: Is the DOI index equal to R? In this case, could you either use DOI OR R?**

Yes, the DOI index is equal to R. We replaced the variable R with "DOI".

**33) Line 320, Figure 3: Here, the computation of the DOI analysis appears before the inversion step. From the explanations above, I understood that this analysis was based on the inversion output ("mean… values observed in all SIP images"). Please check and – if necessary – explain in more detail.**

We fully agree and changed the position of the DOI analysis within the work flow of data processing in Fig. 3. Thank you for spotting this error.

**34) Line 345f: "same values" -> "same values of porosity and seismic velocities"**

We adapted line 356 accordingly.

**35) Line 349ff: Obviously, the selection of a constant value of the surface conductivity is empirically useful. Could this selection also be physically sensible/meaningful? Please add a short discussion of the implications of this assumption.**

Due to the assumption of a constant surface conductivity (whether physically meaningful or empirically derived) in the underlying equations of the joint inversion, a homogeneous contribution of the surface conduction to the bulk electrical conductivity is assumed over the whole study area, ignoring variations in surface conductivity for instance due to ice-rich locations or the presence of fine grains and water saturation. We used a value representative for pure ice, where variations due to a mixture of ice, sediment and rock are ignored. The selection of the constant value could be physically meaningful, i.e. representative of the overall sig_surf of the site, but the aforementioned simplification would still be a problem for the proper ice content and porosity estimation, needless to say that we lack the information for our study site. We addressed this issue within the Discussion section.

**36) Line 358ff: How was the saturating fluid prepared (tap water with NaCl)? Please provide some more detail.**

We used diluted tap water to obtain an electrolyte with a distribution of ions that resembles realistic field conditions. We added these details in the Material and Methods section.

**37) Line 268: Actually, this section is rather a "Results and discussion" section, as most of the results are already discussed in some detail. In particular, in this section the text could be shortened by streamlining (in German "straffen") the presentation of results. E.g., lab and field data could be discussed together instead of separately, which leads to many redundancies in the discussion.**

We agree and presented lab and field data together in Section 3.2.

**38) Line 374: "inflection point in electrical resistivity": The depth slices in Figure 4 are not suitable to show an inflection point in the depth profile. Please consider discussing this concept based on depth profile (e.g. at the positions of the two boreholes).**

We thank the reviewer for the suggestion and added as additional subplot in Figure 4 a 1D depth profile with extracted pixel values at the location of the two boreholes. We hope such visualization helps to discern the inflection point.

**39) Line 376: In the Figure, all resistivity values are shown as log10-scaled values, while the text refers to unscaled values. This makes it unnecessarily hard to follow the discussion. Please consider replacing the corresponding labels of the colour bar in the Figure by unscaled values (e.g. 3.75 -> 5620 Ωm) or similar to make the information more easily accessible.**

We agree with the reviewer and replaced the log10-scaled values by unscaled values.

**40) Line 382 / caption to Figure 4: Please add meaning of white contour lines (surface elevation).**

We added a description in the caption of Figure 4.

**41) Line 387: "The baseline SIP imaging results…" please consider adding "along the permanent monitoring profile"**

Thank you for the suggestion, we added the phrase within this sentence.

**42) Line 410 / Figure 5: Please consider numbering photos in (a) from left to right 1-2-3. It is not clear, why the figures are numbered in inverted order. Imaginary resistivity and phase values of 0.5 Hz and 75 should be plotted on the same scale (colour bar limits) to facilitate a direct comparison of these pairs of sections (as discussed in the main text).**

We numbered the pictures in Figure 5a) in the order as suggested by the reviewer and changed the colorbar limits to the same scale.

**43) Line 419: "including warm (…) and phase (…)" consider adding "in the active layer" (this information is given in the Figure´s caption but would help here in the main text).**

We agree and changed the sentence accordingly.

**44) Line 423: "with no" -> "with only small"**

We agree and changed the sentence accordingly.

**45) Line 416: "above" -> "starting at"**

We agree and changed the sentence accordingly.

**46) Line 449 / Figure 6: Can the field data in panel a) be presented in the same fashion as the lab data in panels b)? I.e. could the curves be coloured based on the temperatures measured in the boreholes to facilitate the comparison of field and lab data even more? For understandable reasons (comparison with field data), lab data is only visualized up to 100 Hz although it has been measured up to 45 kHz. Please consider providing the complete spectral information in the supplementary material.**

Since in Figure 6, we show apparent resistivity and phase values instead of inversion results, we prefer not to show a direct correlation to borehole temperatures measured at different depths within the 2 boreholes.

As mentioned above, we provide now the full spectral information of the SIP laboratory measurements in the Appendix.

**47) Line 459: Please provide more information on why and at which frequency such a peak would be expected (e.g. "…which according to… would be expected around … Hz…")**

We addressed this point in lines 484 and 485.

**48) Line 481 / Figure 7: Could the curves for freezing and thawing be combined in one plot? This would make it easier to appreciate the effect of hysteresis between the two directions of temperature change.**

We appreciate this suggestion and changed figure 7 accordingly.

**49) Line 482 / caption of Figure 7: Please provide information on the measurement frequency displayed here.**

We provided the frequency range used to calculate the phase frequency effect and named the frequency of the displayed impedance magnitude.

**50) Line 505 / Figure 8: Is it a fortunate coincidence that the ice-content data and the phase frequency effect data at the borehole position correlate that well (see Figure 13)? Why show**

the good correlation of the two sub-datasets around the borehole here and discuss the limitations separately later in the manuscript? Neither the ice-content nor the phase frequency effect data is limited to the location around the borehole. Regarding the streamlining (shortening) of the text, please consider discussing this correlation only once.

Thank you for this remark, we agree that the correlation is somehow doubled. We therefore removed the correlation subplots f and g from Figure 8 and show the correlation for different ZOIs in Figure 14.

51) **Line 554: Discussing Figure 10c, the authors state "… we also observe at the field-scale lower $\rho'$, $\phi$ and $\phi FE$ during freezing (from October to January) than during thawing (from May to July)…" Figure 10c does not really support this statement. At any given temperature around 0 °C, I observe more low values in warm colours (summer months = thawing) than low values in cold colours (winter months = freezing). Please check!**

We agree with the reviewer that the effect of supercooling with higher values during thawing compared to the freezing period is clearer for $\rho'$ and $\phi$ at 7.5 Hz than for the $\phi FE$. However, even though smaller, the effect can also be observed for the $\phi FE$, where highest values are detected in May ($\phi FE$=0.75) and $\phi FE$ values in October/November are slightly smaller ($\phi FE$=0.6). We adapted the corresponding lines accordingly.

52) **Line 556-559: Redundant: Has already been discussed above.**

We did not discuss the effect of supercooling in the previous chapters. But we agree, that during the presentation of the laboratory freeze thaw cycles, this point is raised for the first time, which is why we shifted the paragraph to section 3.3.

53) **Line 577-595: Please consider discussing possible sources and effects of data error in a separate subsection (i.e. with a separate title).**

We added a new subsection 4.1 "Reliability of SIP monitoring measurements" within the Discussion section where we address EM coupling effects and temporal changes in data error.

54) **Line 596: Delete "so-called"**

We agree and changed the sentence accordingly.

55) **Line 604-606: Redundant.**

We removed the lines from the manuscript.

56) **Line 626: Please define "R".**

We defined R in lines 745-747.

57) **Line 641 / Figure 11: Please check equation in panel b) (see comment on equation (9).**

We removed the equation from Figure 11 and adapted the original Equ. 9 according to comment 20.

58) **Line 701 / Figure 12c): Please consider showing the temperature data only down to 15 m depth to facilitate the comparison with the phase frequency effect data in Figure 12d).**

We agree and changed original Figure 12c) accordingly.

59) **Line 715: It would be helpful for the understanding, if some suitable lab data or modelled data containing the "ice peak" in the frequency range between 1 and 45 kHz was shown earlier in the manuscript (best where it is mentioned/discussed first).**

We added the broadband SIP laboratory data as an additional figure to the Supplementary Material, where we show the full spectrum of measurements collected on a solid rock sample and a loose sediment sample during controlled freeze-thaw cycles between -40 and +20°C. We therein show that the temperature-dependent relaxation behavior of ice with its maximum is expected in the kHz range.

60) **Line 739: "Blacheck" -> "Blaschek"**

We changed the word accordingly.

61) **Line 870: Data availability statement: A link to the mentioned repository should be provided during the review process to allow the reviewers to assess the information provided.**

Thank you for the reminder, we will provide a link to the repository within the review process.

03/07/2023, Matthias Bücker
TU Braunschweig

Institute for Geophysics and extraterrestrial Physics
Braunschweig, Germany

**References**

Binley, A., Slater, L. D., Fukes, M., and Cassiani, G.: Relationship between spectral induced polarization and hydraulic properties of saturated and unsaturated sandstone, Water Resour. Res., 41, W12417, https://doi.org/10.1029/2005WR004202, 2005.

Flores Orozco, A., Bücker, M., Steiner, M., and Malet, J.-P.: Complex-conductivity imaging for the understanding of landslide architecture, Eng. Geol., 243, 241–252, https://doi.org/10.1016/j.enggeo.2018.07.009, 2018.

Ingeman-Nielsen, T.: The effect of electrode contact resistance and capacitive coupling on complex resistivity measurements, SEG Technical Program Expanded Abstracts, 1376-1380, https://doi.org/10.1190/1.2369776, 2006

Kemna, A., Binley, A., Cassiani, G., Niederleithinger, E., Revil, A., Slater, L., Williams, K. H., Orozco, A. F., Haegel, F. H., Hördt, A., Kruschwitz, S., Leroux, V., Titov, K., and Zimmermann, E.: An overview of the spectral induced polarization method for near-surface applications, Near Surf. Geophys., 10, 453–468, https://doi.org/10.3997/1873-0604.2012027, 2012.

Revil, A. and Skold, M.: Salinity dependence of spectral induced polarization in sands and sandstones, Geophys. J. Int., 187, 813– 824, https://doi.org/10.1111/j.1365-246X.2011.05181.x, 2011.

Wang, C., Slater L. D, Extending accurate spectral induced polarization measurements into the kHz range: modelling and removal of errors from interactions between the parasitic capacitive coupling and the sample holder, Geophysical Journal International, Volume 218, Issue 2, August 2019, Pages 895–912, https://doi.org/10.1093/gji/ggz199

Zimmermann, E., Kemna, A., Berwix, J., Glaas, W., and Vereecken, H.: EIT measurement system with high phase accuracy for the imaging of spectral induced polarization properties of soils and sediments, Meas. Sci. Technol., 19, 094010, https://doi.org/10.1088/0957-0233/19/9/094010, 2008.

**SPECTRAL INDUCED POLARIZATION IMAGING TO MONITOR SEASONAL AND ANNUAL DYNAMICS OF FROZEN GROUND AT A MOUNTAIN PERMAFROST SITE IN THE ITALIAN ALPS**

**Theresa Maierhofer et al.**

**Summary**

**The manuscript covers and introduces a new insight into application of the SIP method for cryological research. It shows that the SIP method is sensitive to the temperature, and therefore, temporal changes of the surface, and confirms that monitoring datasets provide valuable information about permafrost evolution.**

**The Introduction, Material and Methods, Discussion and Conclusion are well written. However, in its current format the manuscript is too lengthy and at times the overarching message is lost in the details. It is advisable to attempt to make the Results section further simplified / concise to improve readability, specifically sections 3.2 – 3.5. It would be helpful to highlight clear messages, so the reader is satisfied to have understood the section before moving onto the next. Another important aspect is to highlight the deficiencies / drawbacks of the SIP field measurement system applied here. The system measures up to 225 Hz but (I am assuming) due to coupling effects and higher errors at higher frequencies, data only up to 75 Hz was considered. So, there is a clear need for newer or improved measurement systems that are stable at higher frequencies, specifically for permafrost research dealing with polarisation of ice – which is a topical research area. An overall comment for the figures in the manuscript is that efforts should be made to reduce text crowding and / or remove panels that do not add value to the figure. Consistent colour scales / themes should be used. The results presented here are interesting and exciting, and adds further knowledge to this emerging research field, thank you for sharing this. Please see below for detailed comments on text and figures.**

Thank you very much for your careful reading and the detailed suggestions. We considered each comment and address them in detail in the following text.

We realize now that our Results Section in the original manuscript is too long and important details get lost. We tried to simplify long descriptions by merging and removing repetitions and redundancies. Therefore, we summarized field and laboratory results instead of describing each result separately as in the case of Section 3.2.

We definitely agree with your comment concerning the need for a further discussion of the measurement instrument accuracy and the reliability of the SIP field data at higher frequencies. We therefore add a subsection "Reliability of SIP monitoring measurements" within the Discussion Section of the revised version of the manuscript, where we quantitatively assess the effect of a high sample resistance and electromagnetic fields (i.e., parasitic capacitive coupling (PCC)) on SIP measurements and examine the accuracy of the DAS-1 measurement device that we used within our SIP monitoring measurements. Therefore, we built an electric circuit using commercial resistors and capacitors with known values (see new Figure 11 in the revised version of the manuscript) as suggested by different authors (Revil and Skold, 2011; Wang and Slater, 2019) and performed measurements with the DAS-1 field device and the Portable Spectral Induced Polarization (PSIP) laboratory unit. Such analysis shows that phase errors due to PCC of the instrumentation or cables can be neglected for frequencies between 0.1 and 75 Hz. Above 100 Hz, phase values strongly deviate from the theoretical response, which is why we did not include these measurement frequencies in our SIP monitoring analysis. If sample resistances exceed $1\ k\Omega$, as could be the case for winter measurements, phase errors increase leading to a high data uncertainty. Nonetheless, freeze-thaw transitions are well-resolved and reveal

consistent seasonal trends, hence, our analysis focuses on the summer period as well as the freezing and thawing period. For a detailed interpretation of winter SIP measurements, future studies should consider the simulation of the effect of high contact impedances of current and potential electrodes, which additionally increases phase errors as pointed out by Ingemann-Nielsen (2006).

Additionally, we tried to reduce subplots, text and panels from different figures that did not add value to the figure (for example Fig. 2, 7, 8 and 11).

We hope that our adaptions answer your comments and we thank you again for your helpful suggestions.

**Line-by-line comments:**
**Abstract**
**1. Line 12: Consider reformulation: We investigate the application of spectral induced polarization (SIP) monitoring, to understand seasonal and annual variations of freeze-thaw processes in permafrost, by examining the frequency dependence of subsurface electrical properties. (grammar)**
Thank you for the suggestion, we changed the grammar and phrasing of this sentence accordingly.
**2. Line 15: The SIP imaging results were interpreted in conjunction with complementary seismic and borehole datasets. (grammar).**
We changed the sentence accordingly.
**3. Line 24: Comma after "observed in field data". (grammar)**
We changed the sentence accordingly.
**Page 1**
**4. Line 28: Comma after (Pepin et. al., 2015) (grammar)**
We changed the sentence accordingly.
**Page 2**
**5. Line 31: Consider simplifying to "and water storage capacities" (grammar)**
We agree and simplified the sentence as suggested.
**6. Line 62: Mudler et. al. 2022 discuss that polarisation effects go beyond 1000 Hz.**
We agree and added a sentence about the high-frequency induced polarization measurements with frequencies up to 100 kHz conducted by Mudler et al. (2022).
**Page 3**
**7. Line 70: Change to "broadband". (grammar)**
We changed the word accordingly.
**8. Line 75: Considering changing "but different IP responses." to "but they can be distinguished due to their distinct IP responses." (grammar)**
Thank you very much for this suggestion, we changed the phrasing accordingly.
**9. Line 87: Aren't points 1 and 2 giving the same message/circular loop? Surface conductivities can be determined through SIP to improve ice-content estimation using petrophysical models. The sentence structure is slightly confusing.**
We agree and restructured the sentence.
**10. Line 92: What does signatures mean here?**
We investigate the frequency dependence of the imaging results, the word "signatures" is not needed in this context.
**11. Line 94: Comma after "state" (grammar)**
We changed the sentence accordingly.
**12. Line 95: Consider adding the abbreviation (φFE) as it is being introduced in the main article for the first time.**
We added the abbreviation in line 101.
**Page 4**
**13. Line 98: Unsure about the difference between volumetric water content and unfrozen water content. Can you explain further?**
We added a sentence explaining unfrozen water content and its estimation with ERT in lines 49-51.

**14. Line 99: Perhaps change "sustain" to "evaluate". (grammar)**
We changed the sentence accordingly.
**15. Line 107: Consider changing "Consist of" to "consisting of". (grammar)**
We changed the sentence accordingly.
**16. Line 109: Perhaps a quick explanation of Gelifluction?**
We changed the sentence accordingly.
**Page 5**
**17. Figure 1: The abbreviation "MON" in the bottom image is not explained? I believe this is the ERT/RST profile. Consider indicating this in the caption. There is a blue dot below "MON" is this intentional? Legend says meteorological sensors (plural) but caption says sensor (singular). Change "SIPM the SIP monitoring profile" to "the SIP monitoring profile (SIPM). Overall, it would be good to be consistent that the IDs (e.g., MON, MD, DBH, etc) are indicated in the caption. It helps when jumping back and forth.**
We fully agree and changed the abbreviations and tried to keep all IDs within the text and caption consistent. The dot is the base station of the differential GPS, thank you for the indication, we removed it.
**Page 6**
**18. Section 2.1 describing the field site can be shortened by removing information repeated further down manuscript.**
We admit that the long explanation about surface characteristics is not necessary at this point and is sufficiently described further down manuscript. We shortened Section 2.1 accordingly.
**Page 7**
**19. Figure 2: I'm not sure if panel (c) is necessary here? It is difficult to follow what the image is communicating. Panel (b) is sufficient in showing the depth at which the temperature is 0 degrees Celsius.**
We agree and reduced the number of subplots within Figure 2.
**Page 9**
**20. Line 172: Can you explain why the range 0.1-1000Hz? What does commonly imply?**
Low-frequency domain spectral induced polarization measurements are collected for frequencies < 1 kHz. We added a half-sentence to make it clear.
**Page 10**
**21. Line 225: Does the symbol $\varphi$ here represent porosity? In this case perhaps it's better to represent it with capital $\Phi$ to avoid confusion with phase shift. Also indicate that this term stands for porosity in text.**
Thank you for the indication, we changed the variable and explained it in the main text.
**Page 11**
**22. Line 223: What is $P_{corr}$ here? This has been explained in line 236. Perhaps it can be brought one sentence forward?**
We agree and moved the equation of $P_{corr}$ to the previous chapter, where we introduce the temperature dependence of the electrical conductivity, as this point was also raised from Reviewer 1.
**23. I'm assuming Pore water resistivity is $p_w$. The abbreviation should be added in brackets.**
We added the abbreviation in brackets.
**24. Line 240: Should the term by $S_{0\,w}$ instead? To keep it consistent with the terminology of $P_f$ and $P_o$**
We agree and changed the variable accordingly.
**Page 12**
**25. Line 275: Consider changing corona pandemic to "COVID-19 Pandemic". (grammar)**
We changed the sentence accordingly.
**Page 13**
**26. Line 315: What term does "R" denote?**
R denotes the DOI index, we changed the term accordingly.
**Page 15**
**27. Consider combining section 2.4 and 2.5 as Complementary geophysical datasets to shorten length of the manuscript.**

We combined and shortened the two sections.
**Page 17**
**28. Line 389: Can you explain why the 75 Hz frequency is used at the high frequency, even though measurements were conducted up to 225 Hz?**
Within our new Section 4.1 we show by means of an electric test circuit experiment that phase errors due to coupling of the instrumentation or cables can be neglected for frequencies between 0.1 and 75 Hz. Above 100 Hz, phase errors of 10 mrad occur, which is why we did not include these measurement frequencies in our SIP monitoring analysis.
**29. Line 392: Can you recheck the sentence structure for "In general comparatively… " Please recheck the terms and consider simplification. (grammar)**
We simplified this sentence.
**Page 18**
**30. Line 406: In general, the information contained in these long brackets are hard to follow and hinder the flow of reading. Perhaps consider denoting the graph with symbols and providing this parameter breakdown in a table.**
We agree and added a Table containing the complex resistivity value range observed along the SIP monitoring profile for different substrate classes.
**Page 20**
**31. Consider making section 3.2 more concise.**
We tried to implement this suggestion by presenting laboratory and field data together instead of separately.
**Page 21**
**32. Line 444: Can you explain why the field data shows higher phase angles? I understand that it corresponds to similar surface cover of small amphibolite rocks. So, it should be similar values as that in the lab? This is an important point to discuss and also include in conclusion – does this have further consequences? Is it a drawback of the measurement device or inversion?**
This is an important point, which will be addressed in the revised version.
Laboratory data is less affected by sources of noise limiting field measurements, even small phase errors lead to an increase in the absolute phase values. Additionally, we observe large contrasts along the SIP monitoring profile with large variations due to changes in porosity such as fractures at depth or fine-grained to coarse blocky debris at the surface. These changes play a big role in the inversion of the field data, especially when inverted at different frequencies. It would be interesting to find out in a future campaign about the influence of small heterogeneities, changes in the porosity within the active layer by using small electrode spacings.
**Page 24**
**33. Figure 8 d-e and f-g show the same dataset in different ways. Perhaps only show panels d-e and indicate correlation r2 values in text.**
We agree and reduced the number of subplots.
**Page 27**
**34. Line 546: The highlighted sentences can be condensed such that "the results correspond with those from laboratory, indicating φFE is higher for loose sediment at subfreezing temperatures, and, a lower freezing/melting is observed for the bedrock." The length of the manuscript now starts to feel cumbersome so care should be taken to reduce repetition of information where possible.**
We thank the reviewer for the suggestion and summarized the paragraph accordingly.
**35. Line 554: Should it be "higher" ρ', φ and φFE during freezing?**
No, we observe lower ρ' and φFE values during freezing compared to thawing processes due to the larger amount of unfrozen water upon freezing when supercooling leads to a remaining pore fluid below the freezing point. This can also be observed in our laboratory results in Figure 7. To make it clear, we added an explanation in Section 3.3 where we show this result for the first time.
**Page 28**
**36. Figure 10c: It is recommended to amend figure 10, specifically the location of 10(c) as it's confusing where these measurements are conducted – and how is it linked to 10(a) and 10 (b). For starters you can indicate that measurements were conducted in the debris layer in the caption or**

**somewhere else in the panel itself. Secondly, move the panel away from a. and b. panels to make it obvious to the reader that it's a different type of result that we are viewing.**
We thank the reviewer for this suggestion and changed Figure 10 accordingly.
**Page 29**
**37. Line 597: Consider changing to "use of Phase Frequency Effect (φFE) as a proxy…" (grammar)**
We changed the sentence accordingly.
**Page 30**
**38. Line 615: Can you further explain why the φFE is increasing with increasing temperatures?**
Thank you for spotting this error. We replaced the phrasing to "decreasing temperatures".
**Page 31**
**39. Figure 11: General comment that this figure comes across very crowded. Consider reducing x axis labels and utilizing major / minor grid lines. Secondly, perhaps the text from can be moved to outside / adjacent to the plot? The colour scale should also be consistent and easily to understand for dry and wet conditions. Lastly, the colour scale of the resistivity vs temperature plot should also be consistent throughout. Also, this panel is not explained in the caption or in text. Is there any impact by removing this panel altogether?**
We tried to come along with all suggestions by reducing labels and text within the figure, deleting subplots not necessary and keeping the color scales consistent throughout the whole figure.
This plot shows seasonal changes at different depths, we prefer to keep the colour scale and assign a colour to each depth.
**40. Line 643: Should it say various depths (0.6 – 20 m)?**
Yes, thank you very much, we corrected the depth value.
**Page 36**
**41. Line 761: Please recheck if AL abbreviation has been introduced before**
We introduce the abbreviation now in the Introduction section.
**Page 39**
**42. Line 816: Indicate that these "are the dominant factors controlling the polarisation at the frequencies measured in this study."**
We changed the sentence accordingly.
**43. Line 828: Consider – "can be resolved by SIP through examining changes in the φFE values." (sentence formulation)**
We changed the sentence accordingly.
**END**

**References**

Ingeman-Nielsen, T.: The effect of electrode contact resistance and capacitive coupling on complex resistivity measurements, SEG Technical Program Expanded Abstracts, 1376-1380, https://doi.org/10.1190/1.2369776, 2006

Revil, A. and Skold, M.: Salinity dependence of spectral induced polarization in sands and sandstones, Geophys. J. Int., 187, 813– 824, https://doi.org/10.1111/j.1365-246X.2011.05181.x, 2011.

Wang, C., Slater L. D, Extending accurate spectral induced polarization measurements into the kHz range: modelling and removal of errors from interactions between the parasitic capacitive coupling and the sample holder, Geophysical Journal International, Volume 218, Issue 2, August 2019, Pages 895–912, https://doi.org/10.1093/gji/ggz199

---

## Referee Report (RR1)

Reviewer Recommendation and Comments for Manuscript Number egusphere-2023-671
**SPECTRAL INDUCED POLARIZATION IMAGING TO MONITOR SEASONAL AND ANNUAL DYNAMICS OF FROZEN GROUND AT A MOUNTAIN PERMAFROST SITE IN THE ITALIAN ALPS**

**Theresa Maierhofer et al.**

**Summary**

The authors present an improved version of their interesting and relevant manuscript. They addressed all my general and specific comments on the original version of the manuscript. Specifically, they removed redundancies, included a detailed discussion of the expected broadband response of ice-containing sediment and rock, which improves the understanding of the mechanism underlying the new "phase frequency effect", and provided some additional discussion on the risk of misinterpreting the effect of electromagnetic coupling in terms of ice content.

However, as detailed in my general comments below, there are still parts of the discussion section that remain hard to follow for the reader. More clearly structuring the manuscript (introductory considerations in the introduction section, results in the results section,… ) would improve the clarity and readability of these parts of the manuscript. In addition, the authors might overestimate the scope of the newly added electric circuit experiments, which – from my point of view may only represent a lower bound of the coupling effects to be expected. These issues together with a number of specific comments and technical corrections listed below, should be straightforward to address. Therefore, I recommend accepting the revised manuscript after an additional minor revision.

**General comments**

Structure and clarity of the discussion section

Mainly in the discussion section, there are still some parts of the manuscript, which are hard to follow as they mix literature reviews (introduction), data processing (methods), presentation of data (results) and discussions. In my specific comments below, I provide some suggestions on how these passages could be improved by more clearly structuring the manuscript.

Reference circuit experiment (section 4.1.1)

The new electric circuit experiment provides interesting extra information on the performance of the field equipment in a high-resistance environment. However, strongly suggest moving the entire section to the supplementary material for two reasons:

Structure: An entire experiment should not be presented in the discussion section. In case you decide to leave the study in the manuscript, please consider presenting the experimental method in the methods section, show the results in the results section and discuss the implications in the discussion section.

Scope of the experiment: I doubt that this experiment carried out under laboratory conditions (perfect coupling of current into the test circuit, no significant spatial extension of the layout, etc.) would be able to assess the full error due to the various coupling effects occurring in a real field measurement. It should rather be taken as a lower bound of the expectable error, i.e. the error in the field data might be much higher than this.

**Specific comments and technical corrections**

Line 34: about -> of

Line 50: … the interpretation… are… -> … the interpretation… **is**…

Line 67: … the enhanced polarization response at… -> … the enhanced polarization response **of water ice** at…

Figure 1: If the shaded map comes from TINITALY, why the credit "Google Maps" in panel 1?

Line 137-139: Refer to (Fig. 2b).

Figure 2, caption: Please specify at which point in space the snow height is measured!

Line 154: Add a "." after "(not shown)"

Line 158: "sums are" -> "is"

Line 176: "for low frequencies < 1 kHz" -> this limitation is not necessary here, please consider removing!

Line 189: "to fit a" -> "to reliably fit a"

Line 201: "polarization effect… are…" -> "polarization effect… **is**"

Line 217: Please consider adding a brief explanation of the concept of super cooling!

Line 237, equation (9): In "$S_w$" the S should be upper case.

Line 242: "at relatively low temperatures" -> "at temperatures below 0 °C"

Line 254: Please consider substituting "electronic conduction" by "high electrical conductivity".

Line 265: "These MG" -> "**The** MG"

Line 283-285: In Fig. 3, low current injections are listed as additional filter criterion. Please check and add here or remove from Fig. 3!

Line 300: "in the error parameters" -> please consider adding "(i.e., $\Delta Z$ and $\Delta \phi$)"

Line 301: "…winter and higher…" -> "…winter, which are expected to result in higher…"

Line 315: "the error model" -> "**an** error model"

Line 315: Please consider introducing the error model and its parameters and explaining how the error model parameters are obtained from the NR misfits of magnitude and phase!

Line 320-322: This explanation of the DOI is not clear. What are $m_1$ and $m_2$ for a given inversion result?

Line 334: "was chosen" -> "**were** chosen"; "and 2 m" -> "**at** 2 m"

Line 338: Please define the error model (by stating the equation) – here or further above (see comment on line 315)

Line 356-357: "We used a surface conductivity…" -> "We tested **inversions with** a surface conductivity…"

Line 373: Please also introduce geometry, arrangement and material of the current electrodes!

Line 376: From this description of the experimental setup its seems as if no measures were taken to avoid polarization of potential electrodes by the current through the sample. Most experimental setups remove the potential electrodes from the tested sample to avoid this

effect. Please add a sentence or two to discuss this aspect and explain why this was not needed/possible!

Line 399: "no variation" -> "no **significant** variation"

Figure 5a: Add a blank space between numbers and units (two times "2 m").

Line 410: "absolute phase values" consider adding "(only $\phi_{75}$)"

Line 414-416: At which (approximate) depth is the bedrock being detected? Please add this information to the sentence!

Line 420, table 2: What do the abbreviations AL and PF mean and where exactly can these materials be found along the profile?

Line 459: Fig 7a and 7b do not show freezing and thawing cycles, respectively.

Line 461: "and cooling" -> "and **during** cooling"

Line 464-468: units and "per" should not be type set in italic.

Line 472-474: "Additionally, we observe a lowering of the freezing point of water due to ions being excluded from ice formation and accumulating in the liquid phase". Which specific observation supports this interpretation? Please provide a short explanation/justification for this statement!

Figure 7: Add labels "a)" and "b)" to the panels of the figure! Add unit "(-)" to the phase frequency effect in 7b.

Line 490 and caption to Figure 8: How are the "vertical 1D logs" obtained (extraction from 2D sections or borehole log)? Please provide a brief explanation!

Figure 9: Please consider using the same colors for both the variations of temperature and electrical parameters at the identical depths. This would reduce the legend and make it easier to compare the various time series amongst each other.

Line 616: "and unfrozen" -> "and **the** unfrozen"

Line 621-622: "Supercooling… at the same temperatures…" This is discussed in the context of Figure 12, which does not contain temperature information. Please detail!

Line 634-654: This section is quite confusing: It introduces complicated concepts and approaches used by other authors (which do not directly link to the section title "Temporal variability of the phase-frequency effect and unfrozen water content") but finally reaches the conclusion that these concepts and approaches cannot be applied to the present data set.

In order to improve the clarity of the (still lengthy) manuscript, I strongly suggest removing this section.

Figure 12:

- c) Add unit "(-)" to UWC on vertical axis
- c) What is "S"? Please explain in the caption or remove from vertical axis!
- Please consider stretching the legend over the entire width of the figure to reduce the large white space.
- Legend: "20m" -> "20 m"

Line 671: Figure 12 does not show ice-content data. Please check and eventually rephrase!

Line 671: Actually, only Figure 8d shows a clear relation between $\phi FE$ and the PJI ice content. Figures 8a and 8c do not show a clear relation between these two parameters.

Line 672: "the proposed parameter" -> "$\phi FE$"

Line 691-692: Fig. 13a and 13d show $\phi FE$, not Fig. 13b and 13c. Please check!

Figure 13:

- a) Add labels "z (m)" to all vertical axes.
- a) Add unit "(-)" to phase frequency effect on horizontal axis.
- b) Add unit "(-)" to phase frequency effect difference on horizontal axis.
- c) Make sure the legend does not mask the data!
- Caption: "difference", "Aug22" and "Aug20" should not be set in italic.

Line 719-721: Please consider breaking down this confusing sentence into clear sentences.

Line 721-733: Please consider moving this part to (or merging this part with the corresponding part of) the introduction! There is no direct link with the section title "Comparison of the phase frequency effect and PJI ice and water content estimations".

Lines 733-738: These practical considerations do also not have any direct link to the section title but rather describe the research gap addressed by this study (move to introduction?).

Line 747-765: These are rather considerations regarding the inversion approach. Please consider discussion these inversion-related aspects in a separate subsection to help improving the clarity of the manuscript!

Line 760: "int" -> "in"

Line 766: Which particular observation in Figure 14 "evidences an over-estimation of the ice-content in the active layer through the PJI"? Please detail!

Line 787: "… due to changes in porosity such as fractures at depth…" As this manuscript does not present any data on fractures etc., this statement remains completely unsupported/speculative. Please check and eventually rephrase!

Figure 14:

- Upper panel: To which date does the resented phase frequency effect data correspond?
- Lower panels: The legends should not mask any of the data points
- Lower panels: What does the symbol size indicate?

Line 800: Empirical petrophysical models linking SIP response and ice content have been proposed earlier (e.g., Zorin and Ageev, 2017).

Line 815: As discussed in the respective general comment, I strongly suggest not to consider the error level observed in the electric circuit experiment as "the accuracy" but rather a lower bound of the error level to be expected in the field.

Line 828: "Cole-Cole" -> "Cole-Cole **model**"

**Additional references**

Zorin, N. and Ageev, D.: Electrical properties of two-component mixtures and their application to high-frequency IP exploration of permafrost, Near Surf Geophys, 15, 603 – 613, https://doi.org/10.3997/1873-0604.2017043, 2017.

19/04/2024, Matthias Bücker
TU Braunschweig
Institute for Geophysics and extraterrestrial Physics
Braunschweig, Germany

---

## Author Response (AR2)

**SPECTRAL INDUCED POLARIZATION IMAGING TO MONITOR SEASONAL AND ANNUAL DYNAMICS OF FROZEN GROUND AT A MOUNTAIN PERMAFROST SITE IN THE ITALIAN ALPS**
**Theresa Maierhofer et al.**

**Summary**
**The authors present an improved version of their interesting and relevant manuscript. They addressed all my general and specific comments on the original version of the manuscript. Specifically, they removed redundancies, included a detailed discussion of the expected broadband response of ice-containing sediment and rock, which improves the understanding of the mechanism underlying the new "phase frequency effect", and provided some additional discussion on the risk of misinterpreting the effect of electromagnetic coupling in terms of ice content.**

**However, as detailed in my general comments below, there are still parts of the discussion section that remain hard to follow for the reader. More clearly structuring the manuscript (introductory considerations in the introduction section, results in the results section,… ) would improve the clarity and readability of these parts of the manuscript. In addition, the authors might overestimate the scope of the newly added electric circuit experiments, which – from my point of view may only represent a lower bound of the coupling effects to be expected. These issues together with a number of specific comments and technical corrections listed below, should be straightforward to address. Therefore, I recommend accepting the revised manuscript after an additional minor revision.**

We thank you very much for the detailed and really helpful comments! We appreciate and considered your suggestions carefully and address them in the following in detail.

We admit that our Discussion Section in the revised manuscript is still too long and tried to structure the manuscript following your recommendations.

Additionally, we agree and decided to move the circuit experiment to the supplementary material and only keep a small section, where we discuss implications to our field monitoring measurements.

**General comments**
**Structure and clarity of the discussion section**
**Mainly in the discussion section, there are still some parts of the manuscript, which are hard to follow as they mix literature reviews (introduction), data processing (methods), presentation of data (results) and discussions. In my specific comments below, I provide some suggestions on how these passages could be improved by more clearly structuring the manuscript.**

We agree and followed the suggestions of the reviewer.

**Reference circuit experiment (section 4.1.1)**
**The new electric circuit experiment provides interesting extra information on the performance of the field equipment in a high-resistance environment. However, strongly suggest moving the entire section to the supplementary material for two reasons:**

**Structure: An entire experiment should not be presented in the discussion section. In case you decide to leave the study in the manuscript, please consider presenting the experimental method in the methods section, show the results in the results section and discuss the implications in the discussion section.**

**Scope of the experiment: I doubt that this experiment carried out under laboratory conditions (perfect coupling of current into the test circuit, no significant spatial extension of the layout, etc.) would be able to assess the full error due to the various coupling effects occurring in a real field measurement. It should rather be taken as a lower bound of the expectable error, i.e. the error in the field data might be much higher than this.**

We thank the reviewer for these helpful suggestions and decided to move the circuit experiment to the supplementary material. We only kept a small section, where we discuss implications to our field monitoring measurements. We also agree that this laboratory experiment represents the lower bound of error that we can expect at the field scale. Hence, we tried to make it clear within the discussion section.

**Specific comments and technical corrections**
1) **Line 34: about -> of**
   We agree with the reviewer and adapted the sentence accordingly.
2) **Line 50: … the interpretation… are… -> … the interpretation… is…**
   We agree with the reviewer and adapted the sentence accordingly.
3) **Line 67: … the enhanced polarization response at… -> … the enhanced polarization response of water ice at…**
   We agree with the reviewer and adapted the sentence accordingly.
4) **Figure 1: If the shaded map comes from TINITALY, why the credit "Google Maps" in panel 1?**
   Thank you for the indication, we removed it from the figure.
5) **Line 137-139: Refer to (Fig. 2b).**
   We agree with the reviewer and adapted the sentence accordingly.
6) **Figure 2, caption: Please specify at which point in space the snow height is measured!**
   We agree with the reviewer and adapted the caption accordingly.
7) **Line 154: Add a "." after "(not shown)"**
   We agree with the reviewer and adapted the sentence accordingly.
8) **Line 158: "sums are" -> "is"**
   We agree with the reviewer and adapted the sentence accordingly.
9) **Line 176: "for low frequencies < 1 kHz" -> this limitation is not necessary here, please consider removing!**
   We agree with the reviewer and adapted the sentence accordingly.
10) **Line 189: "to fit a" -> "to reliably fit a"**
   We agree with the reviewer and adapted the sentence accordingly.
11) **Line 201: "polarization effect… are…" -> "polarization effect… is"**
   We agree with the reviewer and adapted the sentence accordingly.
12) **Line 217: Please consider adding a brief explanation of the concept of super cooling!**
   We added a description of supercooling in lines 481-482.
13) **Line 237, equation (9): In "$S_w$" the S should be upper case.**
   We agree with the reviewer and adapted the equation accordingly.
14) **Line 242: "at relatively low temperatures" -> "at temperatures below 0 °C"**
   We agree with the reviewer and adapted the equation accordingly.
15) **Line 254: Please consider substituting "electronic conduction" by "high electrical conductivity".**
   We agree with the reviewer and adapted the equation accordingly.
16) **Line 265: "These MG" -> "The MG"**
   We agree with the reviewer and adapted the sentence accordingly.
17) **Line 283-285: In Fig. 3, low current injections are listed as additional filter criterion. Please check and add here or remove from Fig. 3!**
   We removed it from Fig 3, as this was not a criterion in our end-filter-crterion.
18) **Line 300: "in the error parameters" -> please consider adding "(i.e., $\Delta Z$ and $\Delta \phi$)"**
   We agree with the reviewer and adapted the sentence accordingly.

19) **Line 301: "…winter and higher…" -> "…winter, which are expected to result in higher…"**
   We agree with the reviewer and adapted the sentence accordingly.

20) **Line 315: "the error model" -> "an error model"**
   We agree with the reviewer and adapted the sentence accordingly.

21) **Line 315: Please consider introducing the error model and its parameters and explaining how the error model parameters are obtained from the NR misfits of magnitude and phase!**
   We added a short paragraph within lines 303-310, where we define the error model applied within our study.

22) **Line 320-322: This explanation of the DOI is not clear. What are $m_1$ and $m_2$ for a given inversion result?**
   We agree and added an explanation within lines 333-339.

23) **Line 334: "was chosen" -> "were chosen"; "and 2 m" -> "at 2 m"**
   We agree with the reviewer and adapted the sentence accordingly.

24) **Line 338: Please define the error model (by stating the equation) – here or further above (see comment on line 315)**
   See comment 22.

25) **Line 356-357: "We used a surface conductivity…" -> "We tested inversions with a surface conductivity…"**
   We agree with the reviewer and adapted the sentence accordingly.

26) **Line 373: Please also introduce geometry, arrangement and material of the current electrodes!**
   We agree and added 2 sentences about the geometry, arrangement and material of the electrodes.

27) **Line 376: From this description of the experimental setup its seems as if no measures were taken to avoid polarization of potential electrodes by the current through the sample. Most experimental setups remove the potential electrodes from the tested sample to avoid this effect. Please add a sentence or two to discuss this aspect and explain why this was not needed/possible!**
   During freezing, we had to ensure a direct contact of the potential electrodes and the measured sample, hence, investigations concerning possible sources of error arising due to the contact between potential electrodes and the tested sample (e.g., Wang and Slater, 2019) and where out of scope of this study. We added a sentence within lines 394-396.

28) **Line 399: "no variation" -> "no significant variation"**
   We agree with the reviewer and adapted the sentence accordingly.

29) **Figure 5a: Add a blank space between numbers and units (two times "2 m").**
   We agree with the reviewer and adapted the figure accordingly.

30) **Line 410: "absolute phase values" consider adding "(only $\phi$75)"**
   We agree with the reviewer and adapted the sentence accordingly.

31) **Line 414-416: At which (approximate) depth is the bedrock being detected? Please add this information to the sentence!**
   We agree with the reviewer and adapted the sentence accordingly.

32) **Line 420, table 2: What do the abbreviations AL and PF mean and where exactly can these materials be found along the profile?**
   We agree with the reviewer that this is misleading and added a description within the table and the description of the table.

33) **Line 459: Fig 7a and 7b do not show freezing and thawing cycles, respectively.**
   We agree with the reviewer and adapted the sentence accordingly.

34) **Line 461: "and cooling" -> "and during cooling"**
   We agree with the reviewer and adapted the sentence accordingly.

35) **Line 464-468: units and "per" should not be type set in italic.**
   We agree with the reviewer and adapted the sentence accordingly.

36) **Line 472-474: "Additionally, we observe a lowering of the freezing point of water due to ions being excluded from ice formation and accumulating in the liquid phase". Which specific observation supports this interpretation? Please provide a short explanation/justification for this statement!**

We added a description of the supporting observation within lines 492-493.

37) **Figure 7: Add labels "a)" and "b)" to the panels of the figure! Add unit "(-)" to the phase frequency effect in 7b.**

We agree with the reviewer and adapted the figure accordingly.

38) **Line 490 and caption to Figure 8: How are the "vertical 1D logs" obtained (extraction from 2D sections or borehole log)? Please provide a brief explanation!**

We agree and clarified the statement within the text and the caption of Figure 8.

39) **Figure 9: Please consider using the same colors for both the variations of temperature and electrical parameters at the identical depths. This would reduce the legend and make it easier to compare the various time series amongst each other.**

We considered all suggestion of the reviewer and adapted the color of the first subplot of Figure 9.

40) **Line 616: "and unfrozen" -> "and the unfrozen"**

We agree with the reviewer and adapted the sentence accordingly.

41) **Line 621-622: "Supercooling… at the same temperatures…" This is discussed in the context of Figure 12, which does not contain temperature information. Please detail!**

We agree with the reviewer and adapted the sentence accordingly.

42) **Line 634-654: This section is quite confusing: It introduces complicated concepts and approaches used by other authors (which do not directly link to the section title "Temporal variability of the phase-frequency effect and unfrozen water content") but finally reaches the conclusion that these concepts and approaches cannot be applied to the present data set. In order to improve the clarity of the (still lengthy) manuscript, I strongly suggest removing this section.**

We can understand the point made by the reviewer and drastically shortened the paragraph.

43) **Figure 12:**
- **c) Add unit "(-)" to UWC on vertical axis**
- **c) What is "S"? Please explain in the caption or remove from vertical axis!**
- **Please consider stretching the legend over the entire width of the figure to reduce the large white space.**
- **Legend: "20m" -> "20 m"**

We agree with the reviewer and adapted the figure accordingly.

44) **Line 671: Figure 12 does not show ice-content data. Please check and eventually rephrase!**

We agree with the reviewer and adapted the sentence accordingly.

45) **Line 671: Actually, only Figure 8d shows a clear relation between $\phi FE$ and the PJI ice content. Figures 8a and 8c do not show a clear relation between these two parameters.**

We agree with the reviewer and adapted the sentence accordingly.

46) **Line 672: "the proposed parameter" -> "$\phi FE$"**

We agree with the reviewer and adapted the sentence accordingly.

47) **Line 691-692: Fig. 13a and 13d show $\phi FE$, not Fig. 13b and 13c. Please check!**

We agree with the reviewer and adapted the sentence accordingly.

48) **Figure 13:**
- **Add labels "z (m)" to all vertical axes.**

- **a) Add unit "(-)" to phase frequency effect on horizontal axis.**
- **b) Add unit "(-)" to phase frequency effect difference on horizontal axis.**
- **c) Make sure the legend does not mask the data!**
- **Caption: "difference", "Aug22" and Aug20" should not be set in italic.**

  We agree with the reviewer and adapted the figure accordingly.

49) **Line 719-721: Please consider breaking down this confusing sentence into clear sentences.**
    We agree and adapted the sentence accordingly.

50) **Line 721-733: Please consider moving this part to (or merging this part with the corresponding part of) the introduction! There is no direct link with the section title "Comparison of the phase frequency effect and PJI ice and water content estimations".**
    We agree and incorporated this part within the Introduction section.

51) **Lines 733-738: These practical considerations do also not have any direct link to the section title but rather describe the research gap addressed by this study (move to introduction?).**
    We agree and incorporated this part within the Introduction section.

52) **Line 747-765: These are rather considerations regarding the inversion approach. Please consider discussion these inversion-related aspects in a separate subsection to help improving the clarity of the manuscript!**
    We prefer to keep these lines within this part of the discussion, as we do not want to open a new chapter.

53) **Line 760: "int" -> "in"**
    We agree with the reviewer and adapted the sentence accordingly.

54) **Line 766: Which particular observation in Figure 14 "evidences an over-estimation of the ice-content in the active layer through the PJI"? Please detail!**
    We added a description within line 766.

55) **Line 787: "… due to changes in porosity such as fractures at depth…" As this manuscript does not present any data on fractures etc., this statement remains completely unsupported/speculative. Please check and eventually rephrase!**
    We agree with the reviewer and adapted the sentence accordingly.

56) **Figure 14:**
- **Upper panel: To which date does the resented phase frequency effect data correspond?**
- **Lower panels: The legends should not mask any of the data points**
- **Lower panels: What does the symbol size indicate?**

  We agree and changed Figure 14 accordingly.

57) **Line 800: Empirical petrophysical models linking SIP response and ice content have been proposed earlier (e.g., Zorin and Ageev, 2017).**
    We agree and added the reference to line 800.

58) **Line 815: As discussed in the respective general comment, I strongly suggest not to consider the error level observed in the electric circuit experiment as "the accuracy" but rather a lower bound of the error level to be expected in the field.**
    We agree with the reviewer and adapted the sentence accordingly.

59) **Line 828: "Cole-Cole" -> "Cole-Cole model"**
    We agree with the reviewer and adapted the sentence accordingly.

**Additional references**

**Zorin, N. and Ageev, D.: Electrical properties of two-component mixtures and their application to high-frequency IP exploration of permafrost, Near Surf Geophys, 15, 603 – 613,**

https://doi.org/10.3997/1873-0604.2017043, 2017.

19/04/2024, Matthias Bücker
TU Braunschweig
Institute for Geophysics and extraterrestrial Physics
Braunschweig, Germany

**Dear Authors, thank you for the revised version of the manuscript.**

**I only have 2 comments which I have detailed below.**

**Figure 7: I understand that additional panels were added with combined freezing & thawing for both loose and solid samples.**
**I suggest removing the initial 4 panels (with green and purple plot curves) as they repeat the same data. The final 2 (new) panels give the complete picture.**

Our intension was to delete the old figure and replace it with the new one. We excuse if there was a mistake and make sure that the new version of the manuscript is correct.

**Line 925: Is "less sources of noise" a realistic explanation for the discrepancy between laboratory and field datasets? I ask because noisy data would mean randomness, however, in these results there seems a trend that both app. resistivities and phase shift trend higher.**
**Secondly, only the apparent resistivities and phase shifts are compared, so the justification of contrasting porosities impacting inversion does not fit here. Perhaps I have misunderstood.**
**I agree that shorter field dipole lengths must be used for a more realistic comparison, i.e., comparable to the laboratory sample size. I also agree that more datasets should be collected to check if this trend continues.**

We agree with the reviewer that our answer was misleading and adapted the paragraph concerning contrasting phase values of laboratory and field measurements. In our opinion the explanation for the discrepancy between laboratory and field datasets consists in larger heterogeneities (according to Fig. 5) and variations in the volumes of investigation at the field scale as well as different sources of contamination in the field data such as electromagnetic coupling effects, which we can reduce to a minimum in the laboratory.

**Many thanks and congratulations again for this exciting research output!**

We again would like to thank you for all your helpful suggestions.

---

## Author Response (AR3)

**Public justification (visible to the public if the article is accepted and published)**:
**Dear Theresa Maierhofer,**

**Thank you for submitting this revised version and responses to referee comments on your manuscript. I find that you have addressed all the issued raised by the two referees and that your manuscript can be published after a couple of technical corrections. Please note a couple of technical comments raised by the editorial support file validation. My one additional comments is that at the end of section 2.5 you have added the sentence "During freezing, we had to ensure a direct contact of the potential electrodes and the measured sample, hence, investigations concerning possible sources of error arising due to the contact between potential electrodes and the tested sample (e.g., Wang and Slater, 2019) and where out of scope of this study.", in response to comment #27 from referee #1. I'm struggling to understand the sentence and how it addresses the referee's concern. Please consider revising this to ensure clarity of the text.**

**I look forward to seeing your manuscript published in TC. Sincerely,**
**Ylva Sjöberg**

Dear Ylva Sjöberg,

Thank you very much for evaluating our manuscript "Spectral Induced Polarization imaging to monitor seasonal and annual dynamics of frozen ground at a mountain permafrost site in the Italian Alps". We appreciate the comprehensive and detailed remarks that helped to improve the quality of the manuscript. In the final version of the manuscript we carefully addressed the technical comments (renumbered figures and tables) as well as the comment concerning section 2.5 (lines 399-401) by improving the formulation and the clarity of the text: "During freezing, we had to ensure a direct contact of the potential electrodes and the measured sample. Investigations concerning possible sources of error arising due to the contact between potential electrodes and the tested sample were out of scope of this study and are further discussed in e.g., Wang and Slater (2019)."

We hope that our edits address your last comments and thank you again.

Best regards,

Theresa Maierhofer, Adrian Flores Orozco, Nathalie Roser, Jonas K. Limbrock, Christin Hilbich, Clemens Moser, Andreas Kemna, Elisabetta Drigo, Umberto Morra di Cella and Christian Hauck